# RuvBL1/2 reduce toxic dipeptide repeat protein burden in multiple models of C9orf72-ALS/FTD

Christopher P Webster[1,2], Bradley Hall[1,2], Olivia M Crossley[1,2], Dana Dauletalina[1,2], Marianne King[1,2], Ya-Hui Lin[1,2], Lydia M Castelli[1,2], Zih-Liang Yang[1,2], Ian Coldicott[1,2], Ergita Kyrgiou-Balli[1,2], Adrian Higginbottom[1,2], Laura Ferraiuolo[1,2], Kurt J De Vos[1,2], Guillaume M Hautbergue[1,2,3], Pamela J Shaw[1,2], Ryan JH West[1,2], Mimoun Azzouz[1,2,4]

**A G4C2 hexanucleotide repeat expansion in *C9orf72* is the most common cause of amyotrophic lateral sclerosis and frontotemporal dementia (C9ALS/FTD). Bidirectional transcription and subsequent repeat-associated non-AUG (RAN) translation of sense and antisense transcripts leads to the formation of five dipeptide repeat (DPR) proteins. These DPRs are toxic in a wide range of cell and animal models. Therefore, decreasing RAN-DPRs may be of therapeutic benefit in the context of C9ALS/FTD. In this study, we found that C9ALS/FTD patients have reduced expression of the AAA+ family members RuvBL1 and RuvBL2, which have both been implicated in aggregate clearance. We report that overexpression of RuvBL1, but to a greater extent RuvBL2, reduced C9orf72-associated DPRs in a range of in vitro systems including cell lines, primary neurons from the C9-500 transgenic mouse model, and patient-derived iPSC motor neurons. In vivo, we further demonstrated that RuvBL2 overexpression and consequent DPR reduction in our *Drosophila* model was sufficient to rescue a number of DPR-related motor phenotypes. Thus, modulating RuvBL levels to reduce DPRs may be of therapeutic potential in C9ALS/FTD.**

## Introduction

Amyotrophic lateral sclerosis (ALS) and frontotemporal dementia (FTD) exist on a clinical spectrum of disease collectively termed ALS/FTD (Ling et al, 2013). The most common genetic cause of ALS/FTD is an intronic hexanucleotide repeat expansion (HRE) of GGGGCC (G4C2) within the first intron of *C9orf72*, referred to as C9ALS/FTD (DeJesus-Hernandez et al, 2011; Renton et al, 2011). The complete molecular mechanism by which this expansion leads to disease is complex. However, three non-mutually exclusive

hypotheses have been proposed: (1) haploinsufficiency of the C9orf72 protein caused by reduced transcription, (2) the formation of abnormal G4C2 and G2C4 repeat RNA foci, and (3) repeat-associated non-AUG (RAN) translation of the repeat into five toxic sense and antisense dipeptide repeat (DPR) proteins; poly-(GA), poly(GR), poly(GP), poly(PA), and poly(PR) (as reviewed in Balendra and Isaacs [2018]). While the exact contribution of each mechanism towards disease pathogenesis is yet to be elucidated, evidence suggests that all three mechanisms could be involved.

We and others have previously described a role of the C9orf72 protein in the regulation of autophagy, with C9orf72 haploinsufficiency leading to reduced autophagy induction and impaired autophagic clearance (Farg et al, 2014; Amick et al, 2016; Sellier et al, 2016; Webster et al, 2016; Aoki et al, 2017). In turn, these autophagy defects appear to exacerbate DPR protein accumulation, promoting DPR-mediated toxicity both in vitro and in vivo (Boivin et al, 2020; Zhu et al, 2020). In addition, post-mortem tissue from C9ALS/FTD patients reveals DPR-positive inclusions are found throughout the brain, co-localising with the autophagy receptor protein SQSTM1/p62 (Cooper-Knock et al, 2012; Mann et al, 2013). Together, these data support a multi-hit model of disease involving both loss and gain of function mechanisms. Taken in isolation, haploinsufficiency of C9orf72 appears insufficient to cause neurodegeneration, instead leading to an autoimmune, inflammatory phenotype in multiple knockout mouse models (Atanasio et al, 2016; Burberry et al, 2016; Sudria-Lopez et al, 2016). However, in addition to its role in autophagy, loss of C9orf72 also disrupts synaptic function, vesicular transport, endocytosis, and in turn, glutamate receptor trafficking (Aoki et al, 2017; Selvaraj et al, 2018; Shi et al, 2018; Bauer et al, 2022a), all of which have the potential to synergize with other C9orf72-associated disease mechanisms, including DPRs, to potentiate neurodegeneration.

While it is probable that all three mechanisms are at play, multiple studies have demonstrated that DPR proteins alone are sufficient to trigger neurodegeneration (Mizielinska et al, 2014; Wen

---

[1]Sheffield Institute for Translational Neuroscience (SITraN), Division of Neuroscience, School of Medicine and Population Health, Faculty of Health, University of Sheffield, Sheffield, UK   [2]Neuroscience Institute, University of Sheffield, Sheffield, UK   [3]Healthy Lifespan Institute (HELSI), University of Sheffield, Sheffield, UK   [4]Gene Therapy Innovation and Manufacturing Centre (GTIMC), Division of Neuroscience, University of Sheffield, Sheffield, UK

Correspondence: c.p.webster@sheffield.ac.uk; m.azzouz@sheffield.ac.uk

et al, 2014; Tran et al, 2015; West et al, 2020). In particular, numerous studies demonstrate that arginine containing DPRs, but also poly-(GA), are likely the most neurotoxic (Kwon et al, 2014; Mizielinska et al, 2014; Wen et al, 2014; Lopez-Gonzalez et al, 2016; Chitiprolu et al, 2018; Saberi et al, 2018; Suzuki et al, 2018; Andrade et al, 2020; Liu et al, 2022). For this reason, methods that alleviate DPR levels may be beneficial in the treatment of C9ALS/FTD.

RuvBL1 and RuvBL2 (also known as RVB1/RVB2, Pontin/Reptin, and TIP49/TIP48) are members of the AAA+ (ATPase associated with diverse cellular activities) family of ATPases. RuvBL1 and RuvBL2 are highly conserved from yeast to mammals, and are paralogous to the bacterial RuvB helicase. Structural analysis via X-ray crystallography and electron microscopy indicates RuvBL1 and RuvBL2 monomers oligomerize into hetero and homo hexameric rings, which can further stack into a double dodecamer ring structure (Matias et al, 2006; Puri et al, 2007; Torreira et al, 2008; Lakomek et al, 2015). The organisation of these oligomeric-hexamers is likely associated with specific functions of the RuvBL1/2 containing complex and is also structurally important for the intrinsic ATPase activity of both RuvBL1 and RuvBL2, which hydrolyse ATP via their conserved Walker A and B motifs (Gorynia et al, 2011; Lakomek et al, 2015). The functions of RuvBL1 and RuvBL2 are diverse and complex. The ATPase activity RuvBL1 and RuvBL2 regulates several cellular processes including nonsense-mediated decay (NMD) (Izumi et al, 2010), DNA damage repair (Kanemaki et al, 1999; Gorynia et al, 2011), mTOR activation (Kim et al, 2013; Shin et al, 2020), chromatin remodelling, and transcriptional regulation (Jha et al, 2013; Zhou et al, 2017; Wang et al, 2022). In many cases, RuvBL1 and RuvBL2 form a scaffold which assists in the organisation of other proteins within a larger macromolecular complex (Matias et al, 2006; Puri et al, 2007; Torreira et al, 2008). This is particularly evident in the chromatin remodelling complexes INO80, SRCAP, and TIP60 and the co-chaperone complex for HSP90, R2TP (reviewed in Dauden et al [2021]). Aside from their nuclear functions, RuvBL1 and RuvBL2 have also been implicated in the control of protein aggregation, specifically assisting in the compartmentalisation of misfolded proteins into the aggresome (Zaarur et al, 2015), a cellular storage compartment for misfolded proteins before their clearance via the autophagy-lysosome pathway (Johnston et al, 1998; Zaarur et al, 2014). More recently, RuvBL1 and RuvBL2 were shown to be involved in the disaggregation of large insoluble aggregates to allow their clearance (Narayanan et al, 2019). By assisting in clearance of aggregates above a critical threshold, RuvBL1 and RuvBL2 are able to maintain protein homeostasis, keeping aggregate formation under control (Narayanan et al, 2019). Loss of RuvBL1 and RuvBL2 prevents the proper formation of the aggresome and also accelerates aggregate accumulation (Zaarur et al, 2015; Narayanan et al, 2019). Given that C9orf72-associated DPRs readily form insoluble aggregates within cells (Mori et al, 2013a, 2013b; Ash et al, 2013; Gendron et al, 2013; Mann et al, 2013; Mackenzie et al, 2014) and are also autophagy substrates (Boivin et al, 2020) potentially delivered to the lysosome via the aggresome, we investigated whether modulating RuvBL1 and RuvBL2 levels could prove beneficial in the removal of these toxic DPR protein species.

Here we report that overexpression of the RuvBL proteins is sufficient to prevent C9orf72-associated DPR formation, which in *Drosophila* is able to rescue a number of motor related phenotypes. These data provide further evidence that modulating DPR levels could be of therapeutic benefit in the development of treatments for C9ALS/FTD.

# Results

## RuvBL1/2 overexpression reduces DPR protein levels in vitro

C9orf72-associated DPR proteins are known as autophagy substrates (Boivin et al, 2020). However, it is well recognised that the DPRs have the potential to form large cytoplasmic aggregates (Mori et al, 2013a, 2013b; Ash et al, 2013; Gendron et al, 2013; Mann et al, 2013; Mackenzie et al, 2014), potentially indicating reduced clearance. The clearance of such aggregates via the autophagy-lysosome pathway is termed aggrephagy (Lamark & Johansen, 2012). Both RuvBL1 and RuvBL2 are implicated in aggrephagy and the disaggregation of larger protein aggregates, including amyloid fibrils, to allow more efficient degradation and clearance (Zaarur et al, 2015; Narayanan et al, 2019). Given these characteristics, we sought to determine if overexpression of RuvBL1/2 could be beneficial in terms of reducing DPR protein levels. To investigate this, we delivered plasmids expressing AUG-driven synthetic, codon-optimised, V5-tagged 100 repeats of poly(GA), poly(GR), or poly(PR) DPRs into HeLa cells co-transfected with plasmids expressing FLAG-tagged RuvBL1 or HA-tagged RuvBL2. A schematic representation of these synthetic DPR constructs is included in Fig S1A–C. Overexpression of FLAG-RuvBL1 and HA-RuvBL2 was confirmed via immunoblot (Fig 1A–D). Because of the challenge of reliably detecting these DPR proteins via standard immunoblot that we experienced previously (Bauer et al, 2022b), V5-DPR levels were analysed via dot-blot. V5-positive signal was present in poly(GA), poly(GR), and poly(PR) transfected cells but absent in the empty vector control transfection indicating these DPRs were readily detected via dot-blot analysis. Overexpression of FLAG-RuvBL1 and HA-RuvBL2 led to a significant reduction in the detectable level of poly(GA) and poly(GR) (Fig 1B and C) but had no effect on poly(PR) levels (Fig 1D). To ensure that the lack of effect on poly(PR) levels was not because of a PR100 saturation, we repeated these poly(PR) experiments, where the level of transfected poly(PR) was titrated down. Overexpression of FLAG-RuvBL1 and HA-RuvBL2 were still unable to affect poly(PR) levels even after a threefold 1:2 serial dilution of total poly(PR) plasmid DNA, suggesting this was not because of saturated levels of poly(PR) (Fig S2A). We next repeated these overexpression experiments with constructs containing 45 uninterrupted sense G4C2 repeats (45xG4C2) with V5-tags in all three frames downstream of the repeats. The orientation of these repeats and the location of the V5-tags is shown (Fig S3). This construct is able to produce all 3 sense DPR proteins via RAN translation (Bauer et al, 2022b; Castelli et al, 2023). Again, overexpression of RuvBL1 and RuvBL2 significantly reduced the detectable levels of V5-DPRs on dot-blot (Fig 1E), indicating RuvBL1/2 can impact the level of DPRs produced via RAN translation. To

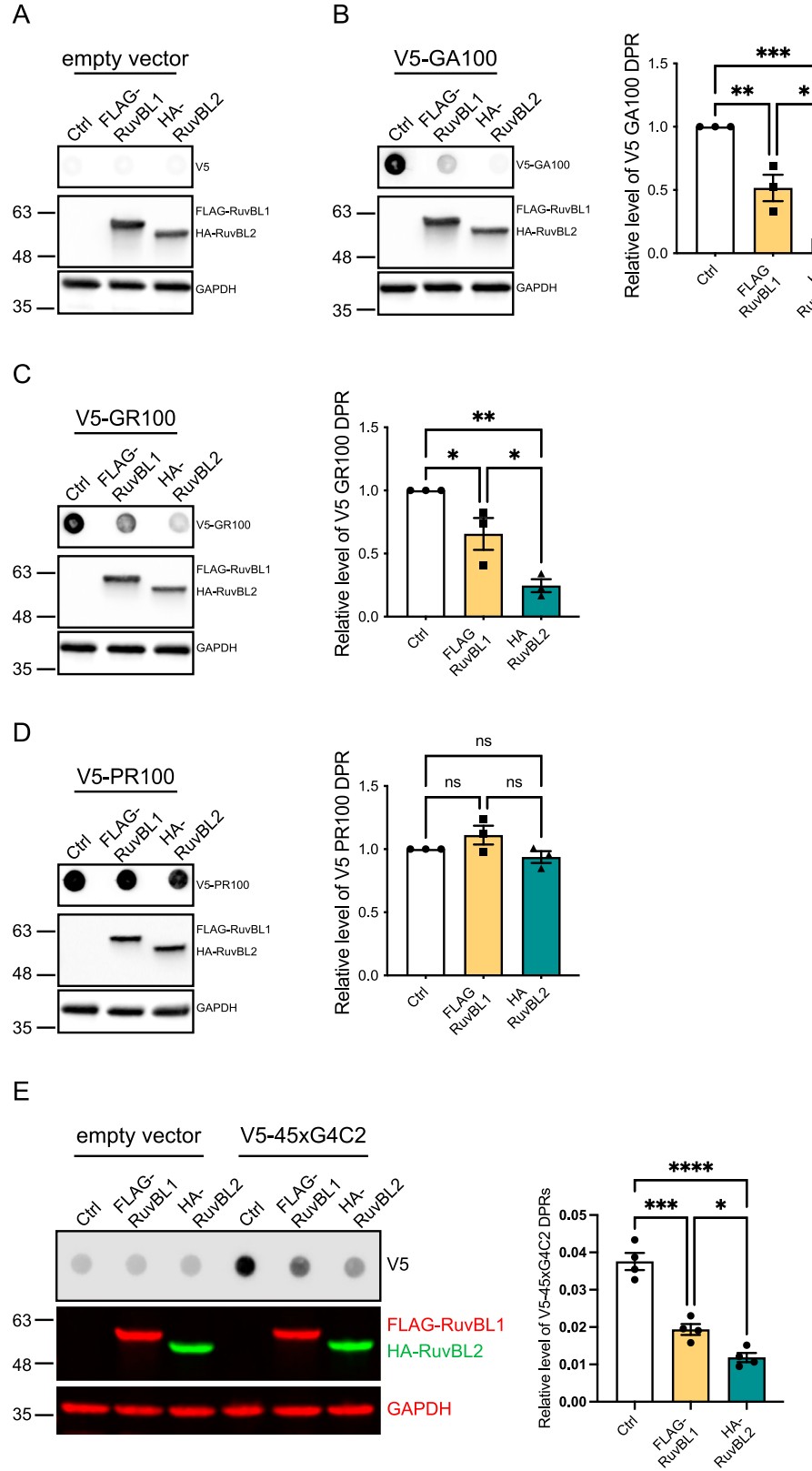

**Figure 1. RuvBL overexpression reduces C9orf72-associated dipeptide repeat (DPR) proteins in vitro.**

**(A, B, C, D)** HeLa cells transfected with empty vector control (Ctrl), FLAG-RuvBL1, or HA-RuvBL2 (A, B, C, D) were co-transfected with empty vector (A) or AUG-driven synthetic codon-optimised V5-tagged 100 repeat poly(GA) (B), poly(GR) (C), or poly(PR) (D) DPR expressing constructs. RuvBL overexpression was confirmed via immunoblot with GAPDH indicating equal loading of samples. Levels of V5-tagged DPRs were determined via dot-blot. DPR levels were normalised to GAPDH and plotted relative to empty vector transfected control (mean ± SEM; one-way ANOVA with Tukey post-test: *$P$ ≤ 0.05, **$P$ ≤ 0.005, ***$P$ ≤ 0.001; **$N$** = 3 independent experiments). **(E)** HeLa cells transfected with empty vector control (ev), FLAG-RuvBL1, or HA-RuvBL2 were co-transfected with empty vector or with 45 uninterrupted sense GGGGCC repeats (45xG4C2) with V5-tags in all three reading frames. RuvBL overexpression was confirmed via immunoblot with GAPDH indicating equal loading of samples. Levels of repeat-associated non-AUG translated V5-DPRs were determined via dot-blot. DPR levels were normalised to GAPDH and plotted relative to empty vector transfected control (mean ± SEM; one-way ANOVA with Tukey post-test: *$P$ ≤ 0.05, ***$P$ ≤ 0.001, ****$P$ ≤ 0.0001; **$N$** = 4 independent experiments).

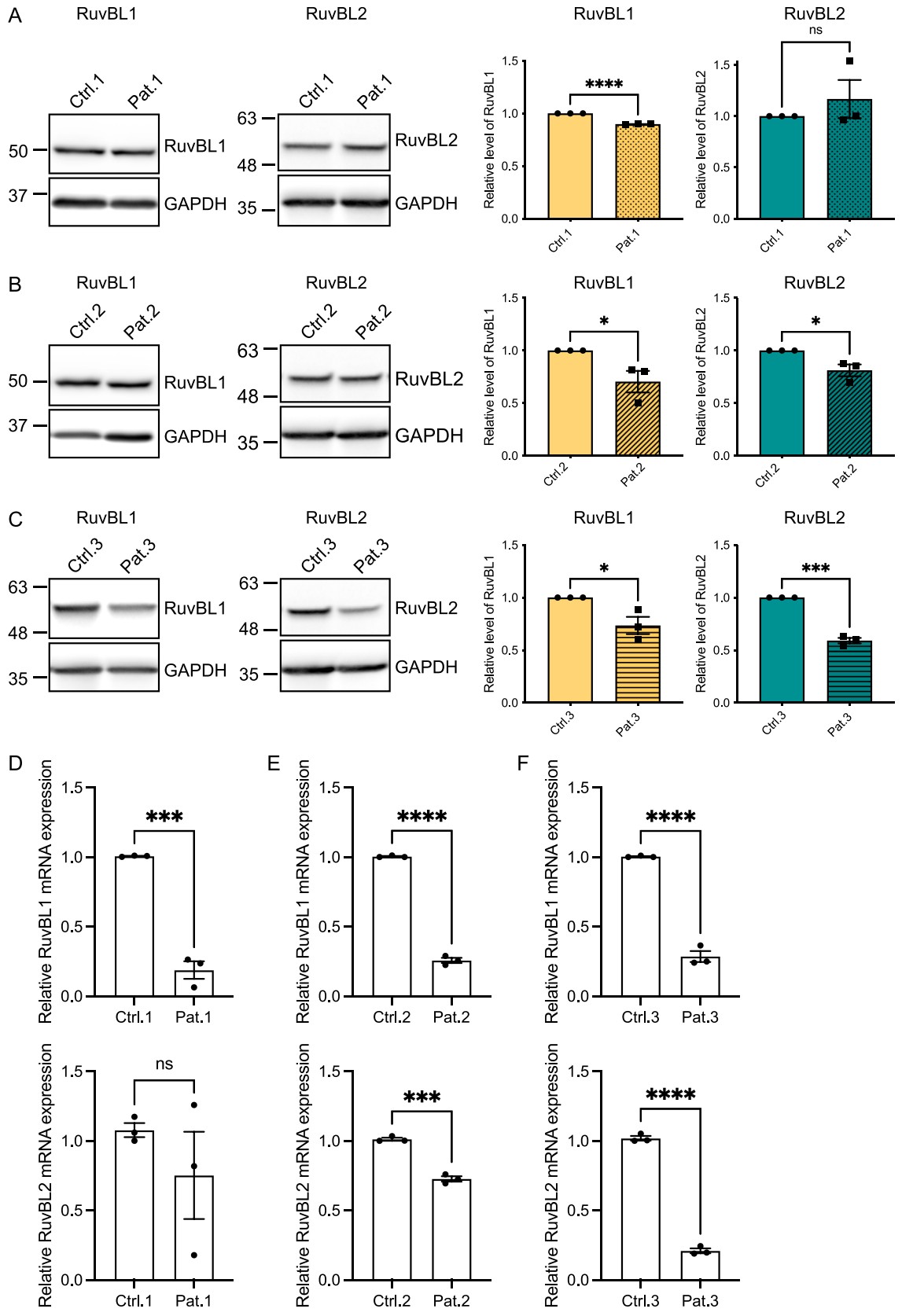

ensure that overexpression of RuvBL1/2 was not affecting protein translation more generally, we repeated these RuvBL1 and RuvBL2 overexpression experiments with a pEGFP-C2 plasmid and measured the levels of EGFP. Overexpression of FLAG-RuvBL1 and HA-RuvBL2 had no significant impact on EGFP levels detected via immunoblot (Fig S2B). To demonstrate the level of RuvBL1/2 overexpression produced by FLAG-RuvBL1 or HA-RuvBL2 transfection, we immunoblotted the above samples with anti-RuvBL1 and anti-RuvBL2 antibodies. Using these antibodies to detect endogenous and exogenous RuvBL1 and RuvBL2 indicated a clear overexpression caused by transfection (Fig S4).

### RuvBL1/2 are differentially expressed in C9orf72 patient cells

The reduced expression of RuvBL1 and RuvBL2 prevents aggresome formation, leading to an increase in cytoplasmic aggregates (Zaarur et al, 2015). Because RuvBL1/2 appeared to reduce DPR levels when overexpressed, we questioned whether C9orf72 patients could have reduced expression of RuvBL1 and RuvBL2. We investigated the level of RuvBL1/2 expression at the mRNA and protein level in C9orf72 patient-derived iAstrocyte cells compared with age and gender matched healthy controls. RuvBL1 protein was found to be significantly lower in all C9orf72-ALS–derived patient cells compared with their controls (Fig 2A–C), whereas RuvBL2 levels were significantly reduced in two of the three patient lines (Fig 2B and C). A similar pattern was observed in the results of the RT-qPCR analysis which indicated a reduced level of expression of *RuvBL1* mRNAs across all patient lines, and a reduced level of *RuvBL2* expression in two of the three patient lines (Fig 2D–F). We next questioned whether it was the presence of RAN-translated DPRs that could be affecting RuvBL1/2 expression in C9orf72 patients. To investigate whether the presence of the repeat expansion could affect RuvBL1/2 protein levels, we transfected our 45 uninterrupted sense G4C2 repeats (45xG4C2) construct into HeLa cells and determined the effect on endogenous RuvBL levels via immunoblot. The presence of this 45xG4C2 repeat has no effect on endogenous RuvBL1 or RuvBL2 (Fig S5A and B).

### Loss of RuvBL1/2 does not increase DPR levels in vitro

The reduced expression of RuvBL proteins in C9orf72 patient cells and the increased DPR reduction when RuvBL1/2 was overexpressed led us to investigate whether loss of RuvBL1 or RuvBL2 would lead to an increase in RAN-translated DPRs. To test this, we reduced RuvBL1 or RuvBL2 expression in HeLa cells using pools of four different siRNAs targeting *RuvBL1* or *RuvBL2* before delivering our RAN-DPR producing construct. HeLa cells were treated with non-targeting control siRNA (siCtrl), RuvBL1-targeting siRNA (siRuvBL1), or RuvBL2-targeting siRNA (siRuvBL2) before transfecting cells with our 45 uninterrupted sense G4C2 repeat (45xG4C2)

construct. Knockdown of RuvBL1 and RuvBL2 was confirmed via immunoblot (Fig S6A and B). We chose to determine the level of poly(GP) DPRs via our highly sensitive, in-house MSD-ELISA. Poly-(GP) DPRs are the most abundant DPR formed from the C9orf72-associated repeat expansion and are also the most readily detected DPR from our 45xG4C2 construct. Poly(GP) DPRs were detected in all cell lysates transfected with V5-45xG4C2 constructs. In these assays, knockdown of RuvBL1 or RuvBL2 had no effect on the level of poly(GP) DPRs detected via MSD-ELISA (Fig S6C).

### RuvBL1/2 overexpression reduces DPRs in primary cortical neurons from C9-500 BAC mice

We next investigated the effect of RuvBL1/2 overexpression on DPR levels in a more physiologically relevant in vitro system. The C9-500 BAC transgenic mouse model expresses a human *C9orf72* gene with ~500 pathogenic G4C2 repeats (Liu et al, 2016). While the behavioural phenotype has been shown to be variable (Mordes et al, 2020; Nguyen et al, 2020), this model does reliably recapitulate a number of the hallmarks specific to C9orf72-related disease including RNA foci and DPR protein production via RAN translation (Mordes et al, 2020; Nguyen et al, 2020). We therefore used this model to investigate the effect of RuvBL1/2 overexpression on DPR proteins at the endogenous, disease relevant, level. Primary cortical neurons were extracted from E16.5 C9-500 BAC-positive embryos and their WT siblings. All embryos were genotyped as described in the Materials and Methods section. To efficiently deliver RuvBL1 and RuvBL2 to these primary cortical neurons, we produced lentiviruses containing FLAG-RuvBL1 or HA-RuvBL2. Our previous analysis of C9-500 BAC cortical neurons via MSD-ELISA has indicated that poly(GA) and poly(PR) DPRs are readily produced in these cells, whereas detection of the other DPR species is more challenging. We therefore investigated whether overexpression of RuvBL1 or RuvBL2 via lentiviral (LV) transduction could reduce poly(GA) and poly(GP) levels in these cells. C9-500 BAC primary cortical neurons were transduced with LV-GFP, LV-RuvBL1, or LV-RuvBL2 at DIV4, before proteins were harvested at DIV10 and LV transduction confirmed by immunoblot (Fig 3A–C). Levels of poly(GA) and poly(GP) were measured by MSD-ELISA (Fig 3D and E). As expected, there was a significant detection of poly(GA) and poly(GP) signals in the C9-500 BAC neurons compared with the WT controls. After transduction with LV-RuvBL1 and LV-RuvBL2, we demonstrated that RuvBL1 and RuvBL2 overexpression significantly reduced poly(GA) DPRs (Fig 3D). However, only transduction with RuvBL2 was able to significantly reduced poly(GP) levels in these assays (Fig 4E). Given that RuvBL levels appeared to be altered in C9orf72 patients, we determined the level of endogenous RuvBL1 and RuvBL2 in these C9-500 BAC transgenic neurons via immunoblot. There was no difference in RuvBL1 levels between C9-500 BAC transgenic neurons and non-transgenic controls (Fig S7A). However, we did observe a

**Figure 2. C9ALS/FTD patient cells have reduced levels of RuvBL proteins.**
**(A, B, C)** RuvBL1 (left immunoblot) and RuvBL2 (right immunoblot) protein levels from 3 C9orf72-ALS/FTD patient iNPC lines and their age and sex-matched controls were determined by immunoblot. Levels of RuvBL1 (left graph) and RuvBL2 (right graph) were normalised to GAPDH and are shown relative to the age and sex-matched control (mean ± SEM; unpaired *t* test: *$P ≤ 0.05$, ***$P ≤ 0.001$, ****$P ≤ 0.0001$, ns, non-significant; $N = 3$ independent experiments). **(D, E, F)** Expression of *RuvBL1* (upper graphs) and *RuvBL2* (lower graphs) transcripts were quantified by RT-qPCR using *18S* as a housekeeping gene (mean ± SEM, N = 3 independent experiments; unpaired *t* test: **$P ≤ 0.005$, ***$P < 0.001$, ****$P < 0.0001$).

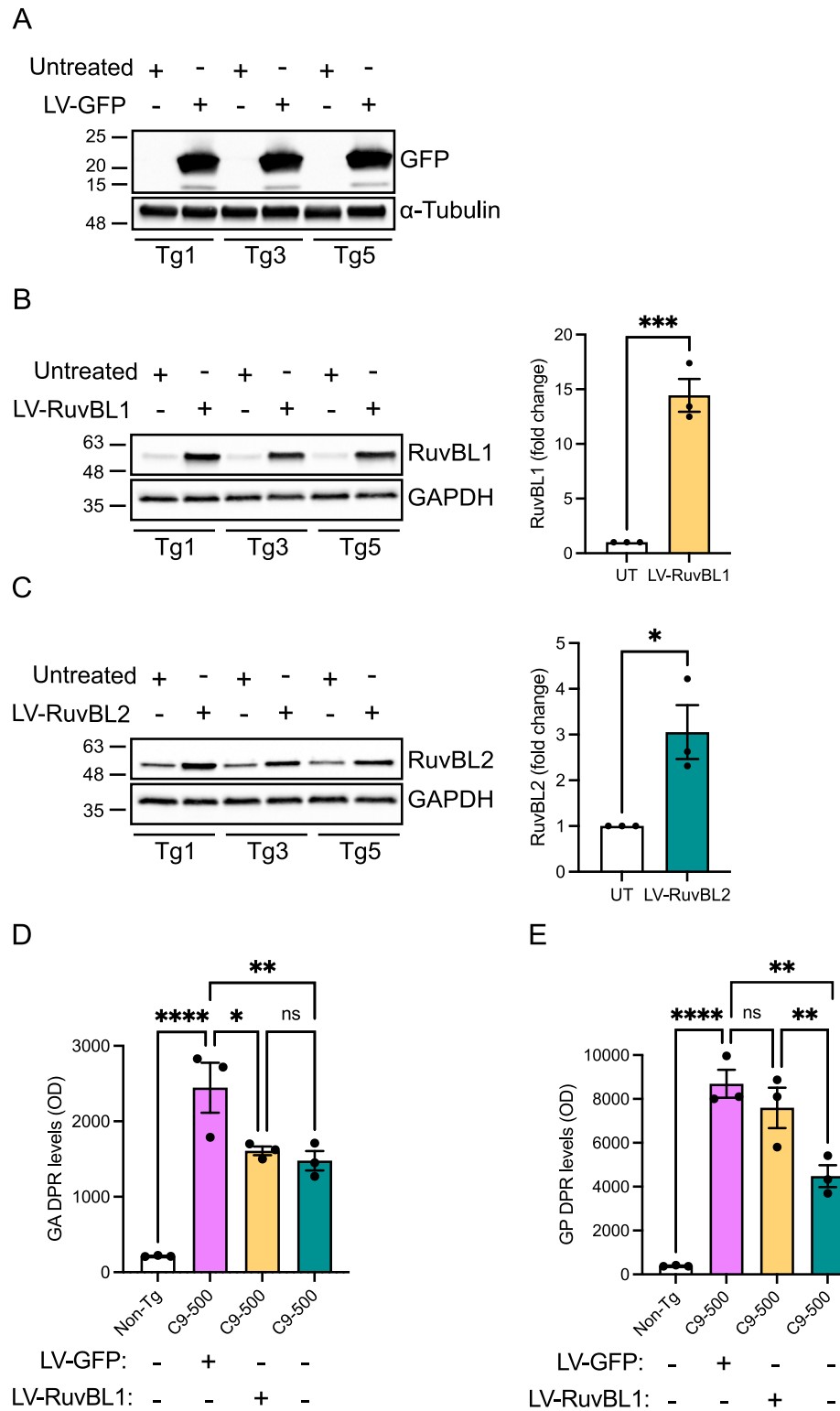

**Figure 3.    Lentiviral transduction with RuvBL2 reduces C9orf72-associated dipeptide repeats (DPRs) in C9-500 BAC primary cortical neurons.**
Primary cortical neurons were extracted from E16.5 WT and C9-500 BAC transgenic (Tg) mouse embryos. At DIV4, Tg neurons were transduced with LV-GFP, LV-RuvBL1, or LV-RuvBL2 at an MOI of 10. At DIV11, proteins were extracted for immunoblot analysis and MSD-ELISA. **(A, B, C)** Transduction and overexpression of GFP (A), RuvBL1 (B), and RuvBL2 (C) was assessed by immunoblot and quantified relative to Tg non-transduced samples. **(D)** Levels of poly(GA) DPRs were assessed by MSD-ELISA (mean ± SEM, N = 3 independent embryos; one-way ANOVA with Tukey post-test: *$P < 0.05$, **$P < 0.01$, ****$P < 0.0001$. ns, non-significant). **(E)** Levels of poly(GP) DPRs were assessed by MSD-ELISA (mean ± SEM, N = 3 independent embryos; one-way ANOVA with Tukey post-test: **$P < 0.01$, ***$P < 0.001$, ****$P < 0.0001$, ns, non-significant).

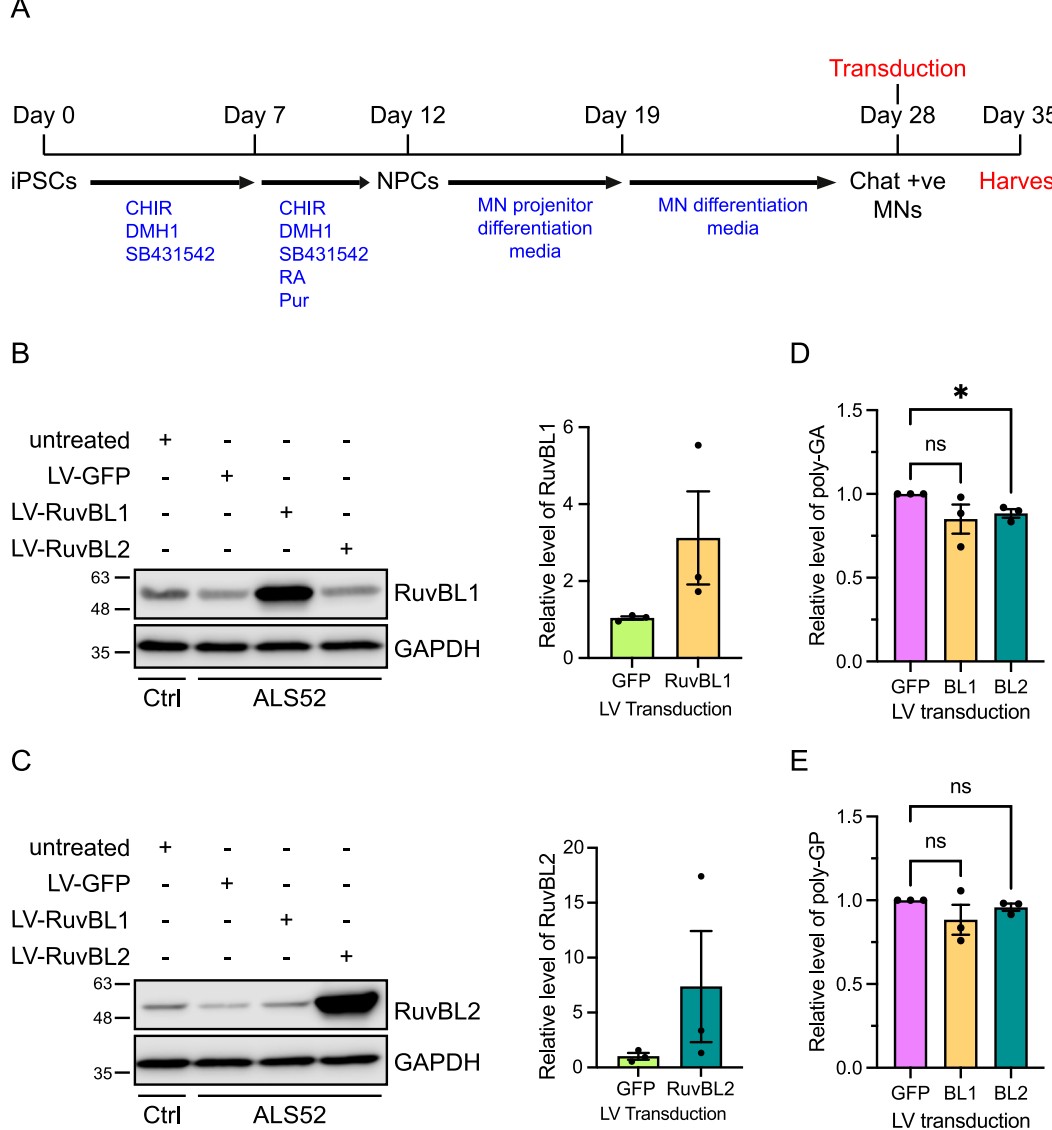

**Figure 4. Lentiviral transduction with RuvBL2 reduces poly(GA) dipeptide repeats (DPRs) in C9orf72 patient iPSC-derived motor neurons.**
(A) A timeline to illustrate the differentiation procedure of the iPSC motor neurons and the timepoint of transduction. iPSC-derived motor neurons from Control (Ctrl: CS14) and C9orf72 patient (ALS-52) were transduced at DIV28 with LV-GFP, LV-RuvBL1, or LV-RuvBL2 at an MOI of 10. 7 d post transduction proteins were extracted for analysis via immunoblot and MSD-ELISA. Transduction and overexpression of RuvBL1. **(B, C)** RA, Retanoic Acid; CHIR, CHIR99021; Pur, Purmophamine; MN, motor neuron (B) and RuvBL2 (C) were confirmed via immunoblot with GAPDH indicating equal loading. Levels of RuvBL1 or RuvBL2 were quantified relative to Tg GFP-transduced samples. **(D)** Levels of poly(GA) DPRs were assessed by MSD-ELISA (mean ± SEM, N = 3 independent experiments; one-way ANOVA with Tukey post-test: *P < 0.05, ns, non-significant). **(E)** Levels of poly(GP) DPRs were assessed by MSD-ELISA (mean ± SEM, N = 3 independent experiments; one-way ANOVA with Tukey post-test: ns, non-significant).

significant decrease in endogenous RuvBL2 levels in transgenic neurons when compared with non-transgenic controls (Fig S7B).

### RuvBL1/2 overexpression reduces poly(GA) DPRs in patient iPSC-derived motor neurons

Having produced lentiviruses capable of transducing primary neuronal cells and overexpressing RuvBL1 and RuvBL2, we then tested these viruses on iPSC motor neurons derived from C9orf72 patients. iPSC MNs were transduced at day 28 of differentiation and

maintained for a further 5 d, at which time proteins were harvested and LV-mediated RuvBL1 and RuvBL2 overexpression, confirmed by immunoblot. In these assays, the level of transduction and, therefore, RuvBL1 and RuvBL2 overexpression was particularly variable (Fig 4A and B). Similar to the C9-500 BAC primary cortical neuron experiments, we measured the levels of poly(GA) and poly(GP) in the protein lysates by MSD-ELISA as these DPRs are the most abundant, leading to more reliable detection and quantification. Overexpression of RuvBL2 led to a significant reduction in the levels of detectable poly(GA) DPRs compared with the GFP-transduced control cells (Fig 4C). However, RuvBL2 had no

significant impact on poly(GP) levels in these assays (Fig 4D). Although LV-RuvBL1 was able to reduce poly(GA) in C9-500 BAC primary neurons, overexpression of RuvBL1 had no impact on poly(GA) or poly(GP) levels in these experiments.

## RuvBL1/2 overexpression reduces DPR proteins in a *Drosophila* model of C9ALS/FTD

After demonstrating that overexpression of RuvBL1 and RuvBL2 were able to reduce DPRs in vitro, including in pathogenically relevant primary and patient cell models, we next turned to in vivo studies. To determine the effect of RuvBL overexpression on DPR production in vivo, we used the previously published *Drosophila* models of C9orf72-related dipeptide repeats, which separately express over 1,000 repeats of either GA, GR, PR, or PA DPRs (West et al, 2020). The *Drosophila* genotypes and genetic crosses used in this study are listed in Tables S1 and S2. *Pontin*, the *Drosophila* orthologue of *RuvBL1* (UAS-Pontin: RuvBL1), *Reptin*, the *Drosophila* orthologue of *RuvBL2* (UAS-Reptin: RuvBL2) or a Gal4 titration control (mKate [UAS-mKate2.CAAX]) were co-expressed pan-neuronally (nSyb-Gal4) with each DPR (UAS-PA[1024]eGFP, UAS-PR[1100]eGFP, UAS-GA[1020]eGFP, UAS-GR[1136]eGFP), or a GFP control (UAS-mCD8-GFP). 7 days post-eclosion (DPE) proteins were extracted from fly heads, and levels of each DPR measured by MSD-ELISA to accurately assess changes in DPR levels between groups. Co-expression of Pontin with GA and GR had no effect on detectable DPR levels (Fig 5A and B) but was able to significantly reduce poly(PR) and poly(PA) levels (Fig 5C and D). Although co-expression of Reptin had no effect on GA levels (Fig 5A), Reptin co-expression significantly reduced the levels of poly(GR), poly(PR), and poly(PA) in these flies (Fig 5B–D). Again, as seen with our previous experiments, the strongest effect in terms of DPR reduction and DPRs affected was observed with co-expression of Reptin, the orthologue of RuvBL2.

## Reptin co-expression rescues GR(1000)-, PR(1000)-, and PA(1000)-associated motor phenotypes in *Drosophila*

Pan-neuronal expression of GR(1000), PR(1000), and PA(1000) have previously been shown to lead to an impairment of motor function, characterised by reduced climbing ability (West et al, 2020; Bennett et al, 2023). Because co-expression of Reptin (RuvBL2) with these DPRs led to a reduction in their detectable levels, we investigated whether this reduction in DPR levels translated to a rescue of impaired motor function. A schematic to illustrate when each assay was performed post-eclosion is included in Fig S8. We evaluated motor performance in flies pan-neuronally co-expressing UAS-PA(1024)eGFP, UAS-PR(1100)eGFP, UAS-GR(1136)eGFP, or UAS-mCD8-GFP with either UAS-mKate2.CAAX, or UAS-Reptin via a startle-induced negative geotaxis assay at 7 and 14 DPE. Pan-neuronal expression of GFP when co-expressed with mKate or Reptin did not show a decrease in climbing from 7 to 14 DPE (Fig 6A). In these assays, pan-neuronal expression of PA(1000) and GR(1000) led to a significant decrease in vertical climbing distance from 7 to 14 DPE in the mKate co-expressing groups (Fig 6B and C). However, in flies co-expressing Reptin (RuvBL2), no significant decrease in vertical climbing was observed (Fig 6B and C), with Reptin

significantly rescuing the vertical climbing distance in PA(1000) expressing flies. These data suggest co-expression of Reptin was sufficient to rescue this progressive motor phenotype associated with PA(1000) and GR(1000) expression. Although PR expressing flies did not exhibit a progressive reduction in climbing between 7 and 14 DPE, co-expression of Reptin in these flies did lead to a significant increase in climbing distance at both 7 and 14 DPE (Fig 6D).

We next assessed motor function by means of the *Drosophila* Activity Monitoring system, an assessment of fly activity which measures the number of "moves" or infrared beam crosses in a 24-h period. This provides a readout of basal activity in the absence of a startle stimulus. Flies from each genotype were analysed at 14 DPE over 24 h. The average moves per hour in each group are plotted in Fig 6E. Flies co-expressing mKate with PA(1000), GR(1000), or PR(1000) displayed a significant reduction in activity compared with the GFP control (Fig 6E and F). Co-expression of Reptin had no effect on the activity of GFP expressing flies and was also unable to rescue the activity defect caused by PA(1000) expression. However, co-expression of Reptin did lead to a partial rescue of the activity defect seen in GR(1000) and PR(1000) expressing flies, as measured by the significant increases in total beam crosses measured over 24 h (Fig 6F). To assess whether these differences were indeed related to motor function, we further analysed the activity monitor data to assess sleep patterns in these flies. Sleep was defined as a temporary stationary period of more than 5 min, as described previously (Silva et al, 2022). Using the Rtivity software to measure total time sleeping over the 24-h period, we discovered that both GR(1000) and PR(1000) flies sleep more during daylight hours (Fig 6G). Co-expression of Reptin was able to rescue this defect in PR(1000) flies and partially rescue in the GR(1000) flies (Fig 6G). No difference was observed in the PA(1000) flies. Taking these results together, these data indicate that increased expression of Reptin (RuvBL2) can rescue the motor deficits caused by the expression of C9orf72-associated DPRs of pathogenic lengths in an in vivo *Drosophila* model of C9orf72-related neurodegeneration.

## RuvBL1/2 overexpression slows the rate of DPR production by affecting repeat RNA levels

The RuvBL proteins have previously been implicated in processing of amyloid fibres and the formation of the aggresome (Zaarur et al, 2015), the subcellular compartmentalisation of misfolded or aggregate prone proteins destined for clearance by the autophagy/lysosome system (Fortun et al, 2003; Iwata et al, 2005). Furthermore, they are also involved in the disassembly of large protein aggregates, acting in a surveillance system to recognise and enhance the clearance of protein aggregates above a critical size threshold (Narayanan et al, 2019). Given that DPRs are aggregate prone proteins capable of forming large insoluble condensates within cells, we assessed whether the effect of RuvBL overexpression on DPR levels was because of an increased rate of clearance. Having already seen a significant reduction in V5-sense DPR levels 48 h after co-transfection with RuvBL1 and RuvBL2 (Fig 1E), we assessed how V5-sense DPR levels changed over time in RuvBL1 or RuvBL2 overexpressing cells after the inhibition of protein translation with cycloheximide (CHX). HeLa cells transfected 24 h previously with

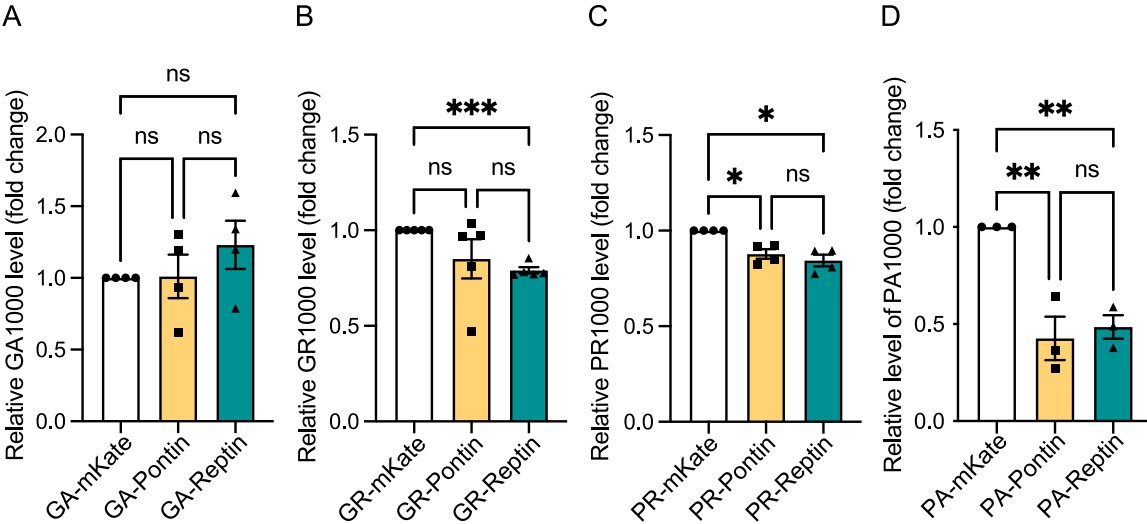

**Figure 5. RuvBL co-expression decreases dipeptide repeat levels in a *Drosophila* model of C9ALS/FTD.**
**(A, B, C, D)** *Drosophila* pan-neuronally (nSyb-Gal4) co-expressing UAS-mKate2.CAAX (mKate), UAS-Pontin (Pontin), or UAS-Reptin (Reptin) with either UAS-GA(1020)eGFP (GA1000) (A), UAS-GR(1136)eGFP (GR1000) (B), UAS-PR(1100)eGFP (PR1000) (C), or UAS-PA(1024)eGFP (P1000) (D) were aged to 7 DPE before heads were taken for protein extraction. Dipeptide repeats were analysed by MSD-ELISA and are presented relative to the mKate control (mean ± SEM, N = 3 [PA1000], 4 [GA1000 and PR1000], or 5 [GR1000] independent experiments; one-way ANOVA with Tukey post-test: *$P < 0.05$, **$P < 0.01$, ***$P < 0.001$, ns, non-significant).

control, FLAG-RuvBL1, or HA-RuvBL2 plasmids were transfected with V5-sense DPR plasmids and protein translation inhibited with CHX 8 h post transfection. RuvBL1 and RuvBL2 overexpression was confirmed by immunoblot (Fig 7A). Cyclin D1, having a short half-life because of rapid turnover, was used as an indicator of translational inhibition and protein clearance (Fig 7A). In these assays, we again determined poly(GP) levels via MSD-ELISA to give the most accurate measure of total DPR proteins (Fig 7B). We discovered that after translational inhibition with CHX, the total level of poly(GP) DPR remained stable and showed no significant level of clearance over the 24-h period studied (Fig 7B). Furthermore, it appeared that the presence of RuvBL1/2 was not able to affect DPR levels when protein translation was inhibited, contrary to what we observed previously in the absence of translational inhibitors (Fig 1A–E). Given that these DPRs were inherently stable with a long protein half-life, these results indicated that RuvBL1 and RuvBL2 were possibly affecting DPR translation and the rate of DPR production rather than clearance. We therefore followed the rate of poly(GP) DPR production from the sense construct via MSD-ELISA immediately following plasmid transfection. Sense plasmids were delivered to HeLa cells which had been transfected 24 h previously with control, FLAG-RuvBL1 or HA-RuvBL2 plasmids. Proteins were then harvested at a range of time points over the next 24 h and poly(GP) levels determined via MSD-ELISA. RuvBL1/2 protein overexpression was confirmed by immunoblot (Fig 7C). The levels of poly(GP) at each time point are shown in Fig 7D. The presence of RuvBL1 and RuvBL2 appeared to slow the rate of DPR production and, indeed, 24-h post DPR transfection there was significantly less poly(GP) in RuvBL2 overexpressing cells compared with control (Fig 7E).

The RuvBL proteins have been shown to regulate transcription coupled translation, and mRNA translatability during glucose starvation and cellular stress (Chen et al, 2022), yet interaction with the translational machinery and ribosome has not been studied. To

further characterise the role of RuvBL1 and RuvBL2 in translation, we investigated whether RuvBL1 or RuvBL2 were able to interact with the translational machinery. To do this, we explored a possible interaction between the RuvBL proteins and the large ribosomal subunit, the 60S ribosomal protein L10a (RPL10a). HeLa cells were transfected with control plasmid or FLAG-tagged RPL10a, before isolating RPL10a with anti-FLAG antibodies and probing the resulting immunoprecipitate for RuvBL1 and RuvBL2. Endogenous RuvBL1, and to a greater extent, RuvBL2 were found to specifically co-immunoprecipitate with RPL10a (Fig 7F and G), suggesting a possible interaction between RuvBL1/2 and this component of the large 60S ribosomal subunit.

Of course, differences in rate of production could also be because of reduced availability of mRNA and reduced transcription rather than translation. Given the dual roles of RuvBL proteins in transcription coupled translation, their interactions with mRNA's and transcriptional initiation (Chen et al, 2022; Wang et al, 2022), in parallel to these translational assays, we also investigated the effect of RuvBL1 and RuvBL2 overexpression on transcription of the sense transcript. Using primers against the sequence downstream of the sense repeat (Illustrated in Fig S9A), we investigated the level of sense DPR transcription via RT-qPCR in cells co-overexpressing FLAG-RuvBL1 or HA-RuvBL2. The presence of RuvBL1/2 overexpression was confirmed via immunoblot (Fig 8A). Normalising to *18S* as a housekeeping gene, we discovered overexpression of RuvBL1 and RuvBL2 had a profound effect on sense DPR transcription (Fig 8B). This effect did not appear to be a reduction in global transcription as *GAPDH* and *C9orf72* expression was unaffected by FLAG-RuvBL1 or HA-RuvBL2 overexpression (Fig 8C and D). FLAG-RuvBL1, HA-RuvBL2, and V5-45xG4C2 repeat expression were all controlled by CMV promoters. To ensure the reduction in V5-45xG4C2 transcripts caused by overexpression of FLAG-RuvBL1 or HA-RuvBL2 was not a consequence of CMV promoter usage, we

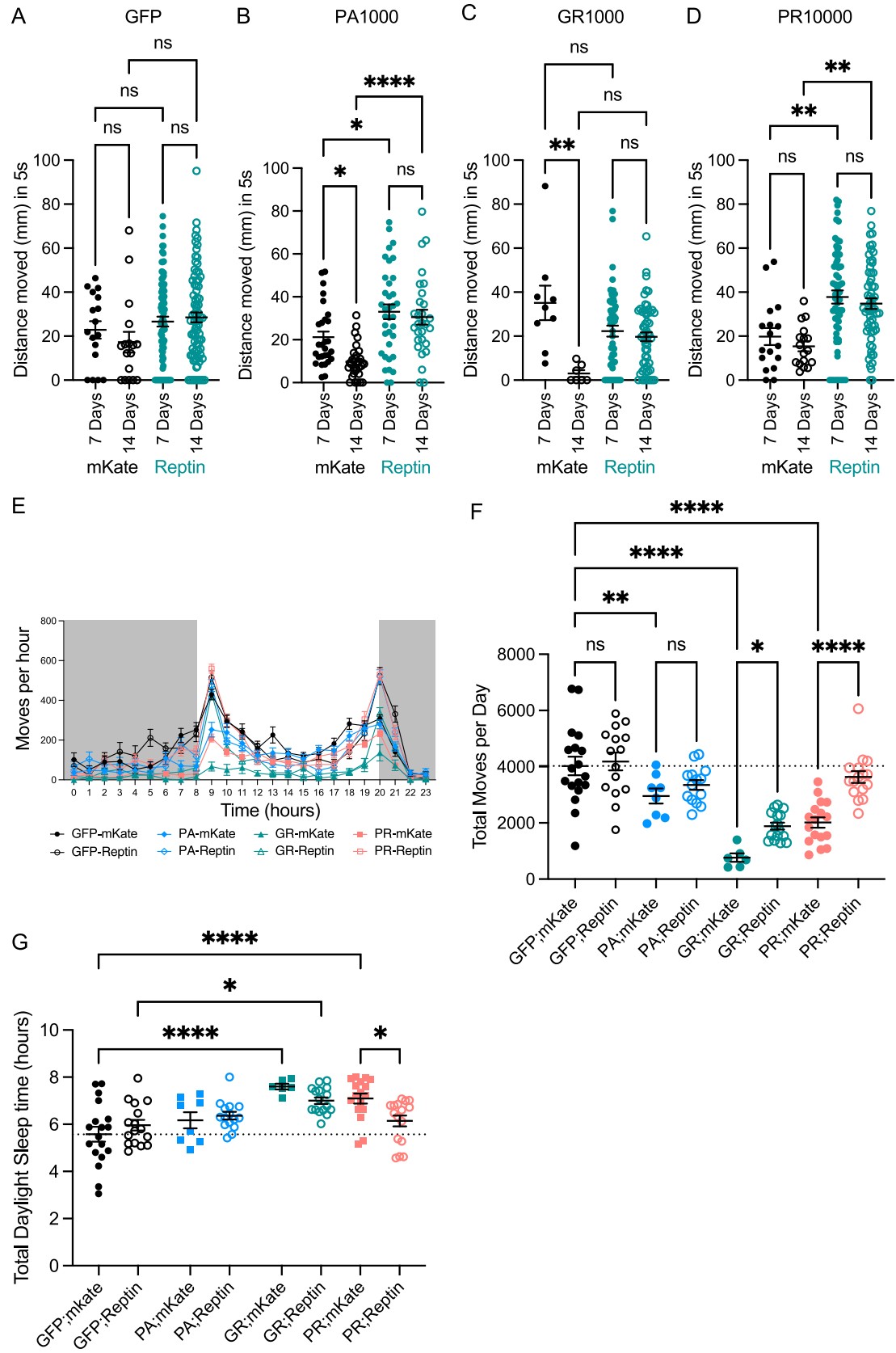

repeated these overexpression experiments with a CMV promoter controlled EGFP expression plasmid, pEGFP-C2. HeLa cells were co-transfected with empty vector control (Ctrl), FLAG-RuvBL1, or HA-RuvBL2 and either empty vector control (Ctrl), pEGFP-C2, or V5-45xG4C2 plasmids. RT-qPCR analysis revealed overexpression of RuvBL1 or RuvBL2 had no effect of *EGFP* transcript levels (Fig S9B), whereas V5-45xG4C2 repeat transcripts were again significantly reduced (Fig S9C). In all these samples, *GAPDH* transcript levels remained unchanged (Fig S9D).

Finally, given that C9orf72 patients displayed reduced expression of RuvBL1/2, we investigated what effect a targeted reduction in RuvBL1/2 expression would have on G4C2 repeat RNA transcript levels. HeLa cells treated with siCtrl, siRuvBL1, or siRuvBL2 were transfected with empty vector control (Ctrl) or V5-45xG4C2 repeats. RT-qPCR analysis revealed RuvBL1-targeting siRNA led to a significant knockdown of both *RuvBL1* and *RuvBL2* transcripts (Fig S10A and B), while RuvBL2-targeting siRNA significantly reduced *RuvBL2* transcripts only (Fig S10B). Knockdown of *RuvBL1* or *RuvBL2* had no significant effect on G4C2 repeat RNA expression in these assays (Fig S10C). Thus, these data indicated that overexpression of RuvBL1 and RuvBL2 are able to reduce transcription of the C9orf72 repeat, leading to reduced DPR translation, which in *Drosophila* is sufficient to rescue a number of neurodegenerative phenotypes.

# Discussion

Here, we demonstrate that overexpression of RuvBL1, but predominantly RuvBL2, are able to reduce C9orf72-associated DPR levels in a range of in vitro models including cell lines, primary neurons, and patient-derived iPSC neuron cells, as well as an in vivo *Drosophila* model of C9orf72-ALS/FTD (Figs 1, 3–5). Furthermore, the reduction in DPRs caused by RuvBL2 overexpression in vivo is able to rescue some of the motor phenotypes associated with DPR expression, including reduced climbing and reduced activity (Fig 6). Given the modifying effect on C9orf72-associated disease pathogenesis, and our discovery that C9orf72 patients display reduced levels of RuvBL1/2 expression (Fig 2), we propose that modulating RuvBL1/2 levels could be beneficial in alleviating DPR-associated disease mechanisms in C9orf72-ALS/FTD. Indeed, when packaged into an AAV9 vector and delivered to primary cortical neurons, we observed RuvBL2 overexpression was again able to reduce poly(GP) DPRs in these cells (data not shown).

RuvBL1 and RuvBL2 are members of the AAA+ (ATPases associated with diverse cellular activities) protein family and are essential components of a number of macromolecular complexes. RuvBL1/2 play critical roles in chromatin remodelling as part of the INO80 (Jonsson et al, 2004; Chen et al, 2011), TIP60 (Shen et al, 2000; Cai et al, 2003), and SRCAP (Shen et al, 2000; Mizuguchi et al, 2004; Cai et al, 2005) complexes, with these having further roles in transcriptional regulation (Makino et al, 1999; Shen et al, 2000; Wang et al, 2022) and DNA damage repair (Kanemaki et al, 1999; Gorynia et al, 2011). Through direct interaction with RPAP3 (Martino et al, 2018; Maurizy et al, 2018), RuvBL1/2 are also involved in the formation of the R2TP complex, a Hsp90 co-chaperone (Boulon et al, 2008, 2010). The scaffold-like function of this R2TP complex, seemingly mediated by RPAP3, may also recruit Hsp70 (Benbahouche et al, 2014; Henri et al, 2018), which is typically involved in the binding and refolding of misfolded proteins and the solubilisation and degradation of aggregated proteins (reviewed in Rosenzweig et al, 2019). By binding and recruiting Hsp70 and Hsp90, the R2TP complex could assist protein client exchange between these two chaperones or even between Hsp70 and RuvBL1/2. Interestingly, RuvBL proteins themselves display chaperone activity (Zaarur et al, 2015; Zhou et al, 2017), specifically in the relation to protein disaggregation and the formation of the aggresome (Zaarur et al, 2015). RuvBL proteins have also recently been implicated in the disassembly of large protein aggregates (Narayanan et al, 2019), a feature previously described for the yeast AAA+ family member, Hsp104 (as reviewed in Shorter and Southworth [2019]), and mammalian Hsp110, which works in conjunction with Hsp70 to promote disaggregation of protein aggregates (Shorter, 2011). RAN translation of the C9orf72 repeat expansion from sense and antisense transcripts gives rise to 5 DPR proteins which form insoluble inclusions within C9ALS/FTD neuronal tissue (Mori et al, 2013a; Ash et al, 2013; Gendron et al, 2013). At least in the case of poly(GA), it appears that soluble oligomers are able to grow into solid insoluble fibrillary aggregates once a critical threshold is reached, similar in nature to amyloid (Brasseur et al, 2020; Marchi et al, 2022). Both Hsp70 and Hsp110 have been implicated in the removal of C9orf72-associated DPR aggregates (Zhang et al, 2021; Liu et al, 2022). Furthermore, given the role of RuvBL in amyloid disaggregation (Zaarur et al, 2015) and the similarity of some DPRs to amyloid fibrils, it could be hypothesised that increasing RuvBL expression promotes clearance either by its own chaperone activity or via client delivery to the Hsp70-Hsp110 disaggregase machinery. Aside from the role of Hsp70, there is further precedent for the involvement of the Hsp70/Hsp90 axis in the disaggregation and clearance of C9orf72-associated DPRs: while Hsp70 and Hsp110 activity appears to promote DPR clearance, pharmacological inhibition and knockdown of Hsp90 also promote DPR clearance and alleviate toxicity (Licata et al, 2022; Lee et al, 2023). Given that inhibition of Hsp90 via geldanamycin induces Hsp70 (Shen et al,

**Figure 6. RuvBL2 co-expression rescues age related motor impairments in *Drosophila* pan-neuronally expressing dipeptide repeats.**
**(A, B, C, D)** The vertical distance climbed 5 s after startle-induced negative geotaxis was recorded in *Drosophila* pan-neuronally (nSyb-Gal4) co-expressing mKate or Reptin with either mCD8-GFP control (A), PA1000 (B), GR1000 (C), or PR1000 (D), at 7 and 14 DPE. The minimum number of flies in any one group was eight (mean ± SEM, flies were from at least three independent crosses per genotype; one-way ANOVA with Tukey post-test: **P < 0.01, ns, non-significant). **(E)** The activity of *Drosophila* pan-neuronally (nSyb-Gal4) co-expressing mKate or Reptin with either mCD8-GFP control, PA1000, GR1000, or PR1000, was assessed at 14 DPE over a 24-h period. The total number moves per hour are presented, with the 12-h dark cycle indicated in grey (mean ± SEM, a minimum of at least six flies were used per group). **(F)** The total number of moves per day per, of each individual animal for each genotype are presented (mean ± SEM, flies were from at least three independent crosses per genotype; one-way ANOVA with Tukey post-test: *P < 0.05, **P < 0.01, ****P < 0.0001). Flies were from at least three independent crosses per genotype. **(G)** The total time (in hours) each individual animal was classified as sleeping during daylight hours are presented for each genotype (mean ± SEM, flies were from at least three independent crosses per genotype; one-way ANOVA with Šidák's multiple comparisons test: *P < 0.05, ****P < 0.0001).

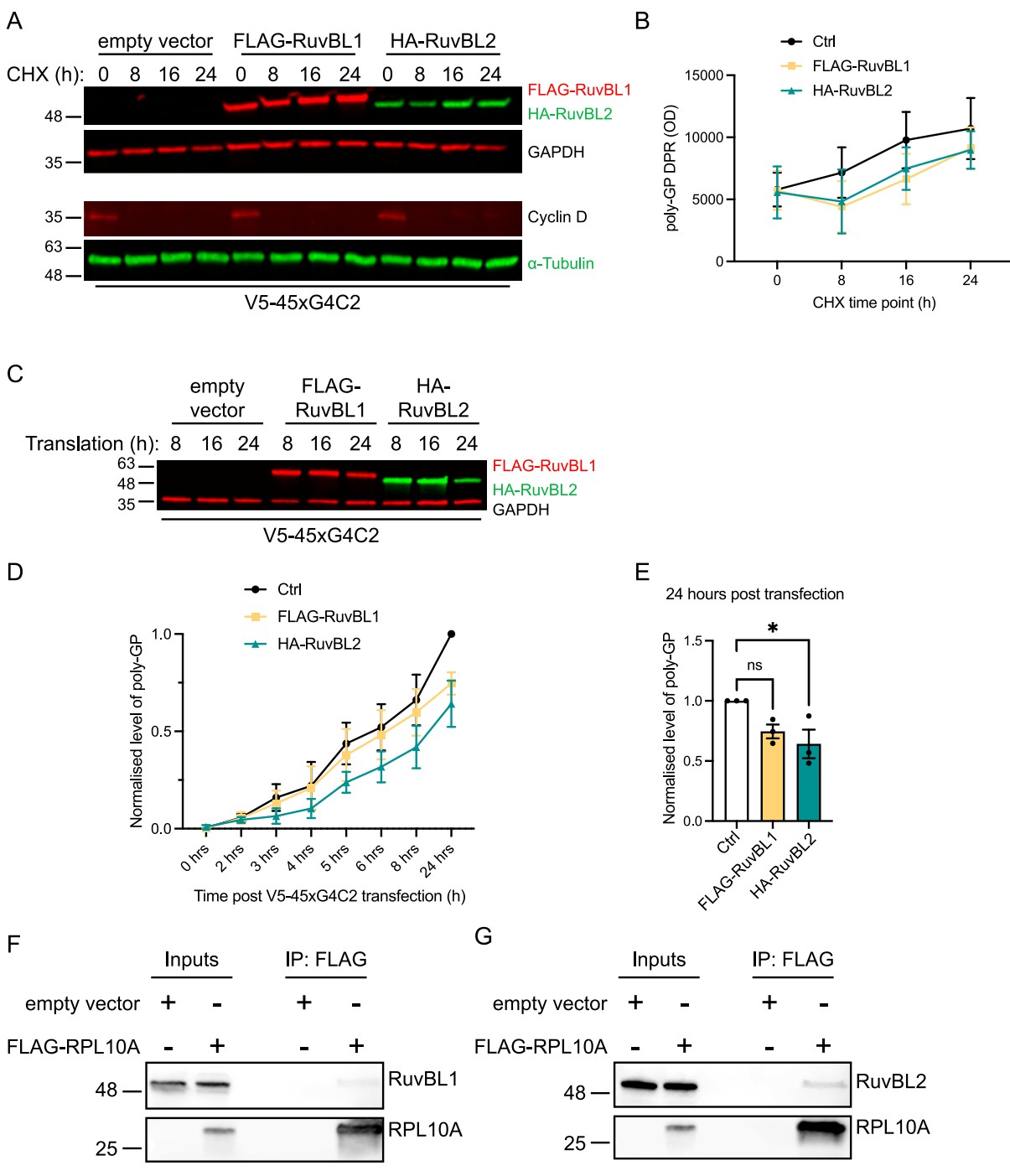

**Figure 7.  RuvBL overexpression slows the rate of dipeptide repeat (DPR) production and interacts with the translational machinery.**
**(A)** HeLa cells transfected with empty vector control (Ctrl), FLAG-RuvBL1, or HA-RuvBL2 were co-transfected with empty vector or with 45 uninterrupted sense GGGGCC repeats (45xG4C2) before treating with cycloheximide (CHX) for the indicated time to block further protein translation. RuvBL overexpression was confirmed via immunoblot with GAPDH indicating equal loading of samples, and cyclin D demonstrating efficacy of the CHX treatment. **(B)** Levels of repeat-associated non-AUG translated poly(GP) DPRs were determined via MSD-ELISA allowing for the monitoring of protein turnover. **(C)** HeLa cells transfected with empty vector control (ev), FLAG-RuvBL1, or HA-RuvBL2 were co-transfected with empty vector or with V5-tagged 45 uninterrupted sense GGGGCC repeats (V5-45xG4C2). Proteins were harvested at the indicated times post transfection to follow rate of production. **(D)** Levels of repeat-associated non-AUG translated poly(GP) DPRs were determined via MSD-ELISA and are presented relative to the empty control transfected 0 h sample. **(E)** The level of poly(GP) DPRs at 8 h post transfection with empty vector control, FLAG-RuvBL1, or HA-RuvBL2. Poly(GP) DPRs are presented relative to the empty vector control (mean ± SEM, N = 3 independent experiments; one-way ANOVA with Tukey post-test: *$P < 0.05$, **$P < 0.01$, ***$P < 0.001$). Lysates from HeLa cells transfected with empty vector control or FLAG-tagged RPL10A were subjected to immunoprecipitation with anti-FLAG antibodies. **(F, G)** Immune pellets were probed for RuvBL1 (F) and RuvBL2 (G) on immunoblot.

A

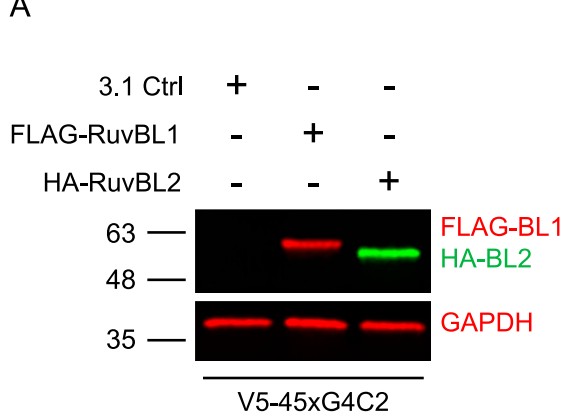

B

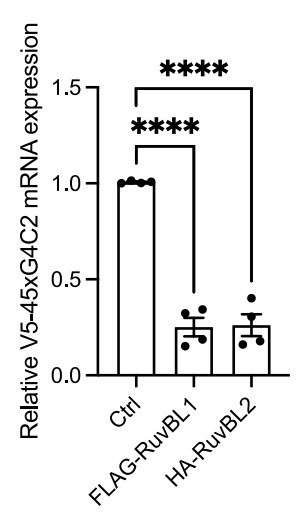

**Figure 8. RuvBL overexpression reduces transcription of C9orf72 sense DNA.**
**(A)** HeLa cells transfected with empty vector control (ev), FLAG-RuvBL1, or HA-RuvBL2 were co-transfected with empty vector or with 45 uninterrupted sense GGGGCC repeats (45xG4C2). RuvBL overexpression was confirmed via immunoblot with GAPDH indicating equal loading of samples. **(B, C, D)** The levels of transcription of the sense repeats (B), *GAPDH* (C), and endogenous *C9orf72* (D) was quantified by RT-qPCR using *18S* as a housekeeping gene (mean ± SEM, N = 4 independent experiments; one-way ANOVA with Dunnett's post-test: ns, non-significant, ****$P$ < 0.0001).

C

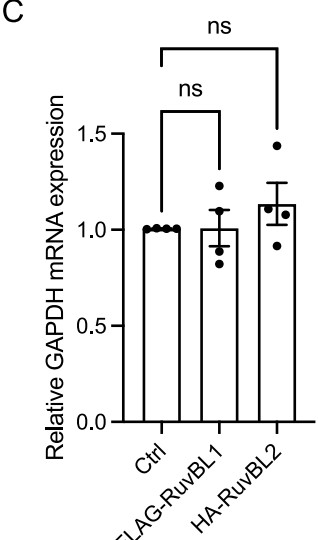

D

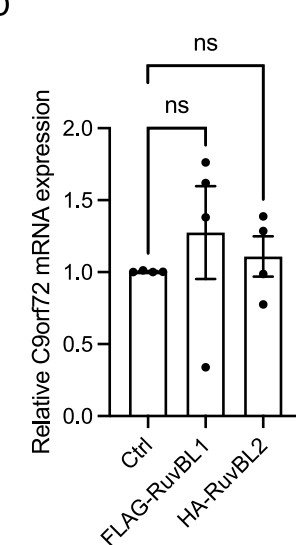

2005), however, it is likely that this effect is also mediated via the Hsp70-Hsp110 machinery. It will be important to fully characterise the relationship between RuvBL overexpression and Hsp70 to gain a more complete understanding of whether these proteins are influencing the Hsp70-Hsp110 disaggregase mechanism, in turn promoting DPR disaggregation and clearance of certain C9orf72-associated DPRs proteins.

Although RuvBL overexpression was able to reduce detectable DPRs and appeared to slow their rate of production (Figs 1 and 7), rather than affecting rate of clearance of poly(GP); it appeared this was because of changes in transcription and/or translation (Fig 8). Recent studies have demonstrated the role of RuvBL proteins in translational repression (Chen et al, 2022). However, RT-qPCR analysis revealed that it was sense DPR transcription that was significantly reduced in RuvBL1- or RuvBL2-transfected cells. The role of RuvBL proteins in transcription and translation is well characterised given their roles in the chromatin remodelling complexes of Ino80, TIP60, and SRCAP. However, because these complexes are associated with chromatin remodelling to bring about gene expression, this function of the RuvBL proteins is typically associated with transcriptional activation. RuvBL2 specifically is associated with the regulation of Pol II clusters and the transcriptional activation of a wide range of genes via its interaction with a diverse array of transcription factors, playing a critical role in global transcription (Wang et al, 2022). Indeed, RuvBL proteins are often found to be overexpressed in certain cancers, potentially because of overall transcriptional activation promoting cell proliferation (Mao & Houry, 2017). In our hands, RuvBL overexpression led to a dramatic reduction in C9orf72 sense expression, while *GAPDH*, *C9orf72*, and *EGFP* transcription remained unchanged (Figs 8 and S9), indicating a specific effect on C9orf72 G4C2 repeat RNA. RuvBL proteins have previously been shown to repress gene expression via the regulation of transcription factors such as beta-catenin and NF-kB (Bauer et al, 2000; Kim et al, 2005). Our assessment that RuvBL2 overexpression had no effect on DPR stability, turnover, and degradation was only based on our

observations relating to poly(GP) levels. Of course, RuvBL could have differing effects on the other C9orf72-associated DPRs. Having only followed poly(GP) in these assays, it could be possible that RuvBL is effecting stability and clearance of other non-poly(GP) C9orf72-associated DPRs.

Our evidence to date suggests that overexpression of RuvBL is not effecting endogenous *C9orf72* expression (Fig 8D) but is instead affecting repeat RNA levels either by modulating transcription or by altering repeat RNA stability or metabolism. In support of the latter, RuvBL proteins are known to regulate formation of a number of PIKK complexes, and via interaction with SMG-1 may function as part of the NMD pathway (Izumi et al, 2010). Furthermore, the ATPase activity of RuvBL1 and RuvBL2 may be required for the assembly of specific factors necessary for NMD initiation (Lopez-Perrote et al, 2020). Retention of the HRE-containing first intron in *C9orf72* is potentially required for RAN translation to occur (Tran et al, 2015; Niblock et al, 2016). Yet intron retention typically leads to mRNA degradation by NMD (Ge & Porse, 2014). However, the presence of arginine-containing DPR proteins has been shown to inhibit upstream frameshift-1 (UPF-1)–mediated NMD mechanisms in C9orf72 patients, potentially as a result of global translational repression (Xu et al, 2019; Sun et al, 2020). Multiple studies have shown that activation of the NMD pathway, particularly via activation of UPF-1, can modify C9orf72 neurotoxicity through a reduction in DPR levels, leading to neuroprotective effects in several in vitro and in vivo models of C9ALS/FTD (Xu et al, 2019; Ortega et al, 2020; Sun et al, 2020; Zaepfel et al, 2021). In our experiments, C9orf72 patients had reduced expression of RuvBL1/2 (Fig 2) and the presence of a pathogenic length G4C2 repeat in the C9-500 BAC mouse cortical neurons appeared to significantly reduce RuvBL2 protein levels (Fig S7). Given the roles of RuvBL1/2 in NMD and our data showing significant effects on HRE RNA, it is possible that by elevating RuvBL levels one could promote the NMD pathway, similar to the effects seen with eRF1 and UPF-1 overexpression (Ortega et al, 2020) and thus reduce the availability of transcripts able to undergo RAN translation leading to a reduction in C9orf72-associated DPR proteins. However, in our siRNA assays, RuvBL1/2 silencing did not lead to an increase in HRE-containing RNA transcripts or an increase in RAN-translated DPR products, suggesting the observed effects were not exclusively because of modulation of NMD (Figs S6 and S10). RuvBL1/2 ATPase activity appears to regulate molecular complex formation necessary for efficient NMD initiation (Izumi et al, 2010; Lopez-Perrote et al, 2020), and loss of RuvBL1/2 leads to reduced clearance of NMD substrates containing premature termination codons (PTCs), bona fide NMD substrates. The HRE-containing transcripts used in our experiments do not contain PTCs. NMD has also been shown to regulate a number of so-called normal mRNA transcripts (Nogueira et al, 2021). Thus, while loss of RuvBL1/2 could affect metabolism of PTC-containing transcripts, the effect on other NMD substrates is currently unclear. Further studies are required to determine whether overexpression of RuvBL proteins is able to promote NMD and therefore effect levels of *C9orf72* transcripts retaining intron 1. Alternatively, RuvBL1/2 could be affecting transcription of the HRE.

A consistent finding across all our assays was that RuvBL2 overexpression had the most, and in some cases the only, significant effect on DPR reduction. Indeed, in our in vivo *Drosophila* studies, we chose only to focus on the effect of Reptin, the *Drosophila* orthologue of RuvBL2, in terms of phenotypic rescue. This is somewhat surprising but not entirely unprecedented. RuvBL1 and RuvBL2 are known interaction partners, with the crystal structure of the RuvBL1-RuvBL2 complex indicating they exist as heterohexameric rings which stack as a dodecamer (Gorynia et al, 2011). However, it has also been demonstrated that RuvBL1 and RuvBL2 exist in many different oligomeric conformations, including homohexamers (Puri et al, 2007; Niewiarowski et al, 2010). These differences in the structure of the RuvBL1-RuvBL2 complex could potentially indicate varied functions depending on the cellular process, in which they are involved (Nano & Houry, 2013). In certain instances, RuvBL1 and RuvBL2 may even function alone (Magalska et al, 2014; Zaarur et al, 2015), as was seen in our experiments, where RuvBL1 or RuvBL2 were overexpressed independently of the other. Where RuvBL1 and RuvBL2 have also been shown to work independently, they often appear to function antagonistically (Bauer et al, 2000; Kim et al, 2005; Diop et al, 2008). This did not appear to be the case in our study, as RuvBL1 overexpression often showed a similar trend to the overexpression of RuvBL2. Given that endogenous RuvBL1 and RuvBL2 are present in our assays, it will be interesting to determine the exact mechanism by which RuvBL2 overexpression exerts a more pronounced effect on C9orf72-associated DPR levels compared with RuvBL1 and whether this is indeed independent of RuvBL1.

Finally, we and others have previously demonstrated that C9orf72-associated DPRs lead to increased levels of DNA double strand breaks, when also inhibiting the correct DNA damage response (DDR) (Farg et al, 2017; Walker et al, 2017; Andrade et al, 2020). Furthermore, recent studies have indicated that the C9orf72 protein itself is involved in the DDR, with haploinsufficiency of C9orf72 exacerbating the DNA damage caused by poly(GR) DPRs (He et al, 2023). Thus, genome instability appears to contribute to disease pathogenesis in C9orf72-ALS/FTD. As part of the chromatin remodelling complexes, which are necessary for remodelling of the DNA around damage sites, both RuvBL1 and RuvBL2 are implicated in DNA damage repair (Shen et al, 2000; Mizuguchi et al, 2004; Jha et al, 2008). Here again, RuvBL1/2 overexpression could prove beneficial in the context of C9orf72-ALS/FTD, firstly by reducing the DPRs that lead to increased DNA damage and dysfunctional repair and, secondly, by promoting a functional DDR.

While the involvement of RuvBL1/2 in protein disaggregation and clearance was the initial focus of this study, we present here novel data indicating RuvBL1 and RuvBL2 may assist in reducing the pathogenic DPR proteins found in C9orf72-ALS/FTD by modulating the availability of HRE-containing RNA. This modifying effect on DPR levels warrants further investigation, and potentially indicates that modulating RuvBL levels could be beneficial in a C9ALS/FTD setting. This suggestion is supported by our finding that C9ALS/FTD patients may have reduced expression of RuvBL proteins in the first instance. Future work should use these findings to explore potential therapeutic avenues relating to RuvBL1 and RuvBL2.

**Table 1.** Thermal cycling conditions for genotyping PCR.

| Step | Temperature (°C) | Time (s) | Number of cycles |
|---|---|---|---|
| Initial denaturation | 96 | 180 | 1 |
| Denaturation | 94 | 45 | |
| Annealing | 55 | 45 | 32 |
| Elongation | 72 | 60 | |
| Final elongation | 72 | 360 | 1 |

# Materials and Methods

### Plasmids

pCDNA3.1 was used as an empty vector control plasmid (Invitrogen). pEGFP-C2 was used as a CMV promoter control. pCMV3-N-FLAG-RuvBL1 and pCMV3-N-HA-RuvBL2 were purchased from Sino Biological Inc. To incorporate 5′-XhoI sites and 3′-NotI sites onto FLAG-RuvBL1, FLAG-RuvBL1 cDNA was amplified from pCMV3-N-FLAG-RuvBL1 by PCR using 5′-ACGTCTCGAGATGGATTACAAGGATGAC-3′ and 5′-ACGTGCGGCCGCT-TACTTCATGTACTTATC-3′ primers. To incorporate 5′-XhoI sites and 3′-NotI sites onto HA-RuvBL2, HA-RuvBL2 cDNA was amplified from pCMV3-N-HA-RuvBL2 by PCR using 5′-ACGTCTCGAGATGTATCCT TACGACGTG-3′ and 5′-ACTGGCGGCCGCTTAGGAGGTGTCCATGGT-3′ primers. FLAG-RuvBL1 and HA-RuvBL2 were subcloned into a self-inactivating lentiviral (SIN-W-PGK) vector containing the multiple cloning site from pCI-Neo (pLenti-Vos) using the XhoI and NotI restriction sites. AUG-driven synthetic, codon-optimised, V5-tagged 100 repeat poly(GA), poly(GR), or poly(PR) DPR constructs in pCI-Neo were described previously (Bauer et al, 2022b). pcDNA3.1-G4C2×45-3xV5 was described previously (Castelli et al, 2023).

### Cell culture and transfection

HeLa cells were cultured in DMEM (Sigma-Aldrich) supplemented with 10% FBS (Labtech) and 1 mM sodium pyruvate (Sigma-Aldrich) in a humified, 5% $CO_2$ atmosphere at 37°C. Cells were transfected with plasmid DNA using Lipofectamine 2000 (Invitrogen) at a ratio of 2:1 ($\mu$l Lipofectamine 2000: $\mu$g plasmid DNA) according to the manufacturer's instructions. Cells were used for experiments 24–48 h after transfection. HeLa cells were transfected with siRNA using Lipofectamine RNA iMax (Invitrogen) according to the manufacturer's instructions. Cells were DNA transfected 3 d post siRNA transfection and harvested for experimental analysis 4 d post siRNA transfection.

Primary cortical neurons were isolated from E16.5 embryos of C9-500 BAC transgenic mice. Cells were isolated as described previously (Marrone et al, 2022) and cultured on tissue culture plates pre-coated with poly-D-lysine in neurobasal medium supplemented with B27 supplement (Invitrogen), 100 IU/ml penicillin, 100 mg/ml streptomycin, and 2 mM L-glutamine.

### siRNA

Non-targeting control siRNA was purchased from Dharmacon. RuvBL1 and RuvBL2 SMARTpool ON-TARGETplus siRNA's were purchased from Dharmacon. The sequences were as follows: RuvBL1: auaaggguggugaacaagua, gggaaggacagcauugaga, cag-gauaaguacaugaagu, cucaggagcugggguaguaa, RuvBL2: uaa-caaggauugagcgaau, cgcaguacaugaaggagua, gaaacgcaaggguacagaa, gcgagaaagacacgaagca.

### Genotyping of E16.5 mouse embryos

Genotyping was performed on genomic DNA that was extracted from tail tissue taken from the E16.5 embryos after cortical neuron extraction. Genomic DNA was extracted by incubating tail tissue in QuickExtract DNA Extraction Solution at 65°C for 30 min, followed by 4 min at 98°C. Genotyping PCRs were performed in a 25 $\mu$l reaction volume containing both control and transgene primers. Reactions contained 12.5 $\mu$l NEB Quick-Load Taq Master Mix (M0271L; NEB), 100 nM of forward and reverse primers for the control genotyping reaction (Vgll4-F: 5′–TTGGATGGAGAAGGATGGAG-3′; Vgll4-R: 5′–GTCTCCACAAGCCCATGAGT-3′), 200 nM of forward and reverse primers for the transgene genotyping reaction (C9-GT-F: 5′–AGTTGGGTCCATGCTCAACAA-3′; C9-GT-R: 5′–ACTGTTCTAGGTACCGGGCT-3′), and 1 $\mu$l genomic DNA from the QuickExtract DNA Extraction protocol. The thermal profile of the PCR reaction is shown in Table 1. PCR products were resolved on 2% agarose gels in Tris-acetate-EDTA buffer. Control products were visualized at 589 bp. Transgene products were visualized at 314 bp. A single band at 589 bp identified non-transgenics. A band at 314 and 589 bp identified transgenics.

### RNA extraction and RT-qPCR quantification

RNA was extracted from HeLa and iAstrocyte cell pellets using TRIzol reagent (Invitrogen) according to the manufacturer's instructions and resuspended in 20 $\mu$l nuclease-free water. RNA was DNaseI treated (Roche) and quantified using a NanoDrop (NanoDropTechnologies). 2 $\mu$g RNA from whole cell extraction was reverse transcribed into cDNA using M-MLV reverse transcriptase (Invitrogen) according to the manufacturer's instructions. Briefly, 2 $\mu$g RNA was reverse transcribed in a final volume of 20 $\mu$l containing 1 $\mu$l random hexamers, 1 $\mu$l of 10 mM dNTPs, 4 $\mu$l of 5x reverse transcriptase buffer, 2 $\mu$l of 0.1 M DTT, and 1 $\mu$l of M-MLV reverse transcriptase. RT-qPCR was performed using a C1000 Touch thermos Cycler using the CFX96 Real-Time System (Bio-Rad). Samples were amplified in triplicate using the Brilliant III Ultra-Fast SYBR Green QPCR Master Mix (Agilent Technologies) and 250 nM using an initial denaturation step, and 45 cycles of amplification (95°C for 30 s; 60°C for 30 s; 72°C for 1 min) before recording melting curves.

Data were analysed using the Bio-Rad CFX Manager software and relative gene expression determined using the ΔΔCt method, with 18S rRNA used as a reference housekeeping gene.

Primer sequences were as follows: 18S, FW 5′-CGGA-CATCTAGGGCATCAC-3′, REV 5′-GTGGAGCGATTTGTCTGGTT-3′; GAPDH, FW 5′-GGTGGGGCTCATTTGCAGGG-3′, REV 5′-GGGGGCATCAGCA-GAGGGG-3′; C9orf72, FW 5′-GTTGATAGATTAACACATATAATCCGG-3′, REV 5′-AGTAAGCATTGGAATAATACTCTGA-3′; RuvBL1, FW 5′-AGAG-CACTACGAAGACGCAG-3′, REV 5′-TATGACGCCACATGCCTCTC-3′; RuvBL2, FW 5′-AACCGTTACAGCCACAACCA-3′, REV 5′-TTGCGAAGCCTGCC-GAG-3′; 45xG4C2 reporter, FW 5′-GGGCCCTTCGAACAAAAACTC-3′, REV 5′-GGGAGGGGCAAACAACAGAT-3′; EGFP, FW 5′-AAGGGCATCGACTTCAAGG-3′, REV 5′-TGCTTGTCGGCCATGATATAG-3′.

### iNPC production and iAstrocyte differentiation

Skin biopsies were obtained from the forearm of subjects after informed consent, in accordance with guidelines set by the local ethics committee (Study number STH16573, Research Committee reference 12/YH/0330). Fibroblast cell cultures were established in DMEM supplemented with 10% FBS (Labtech), 2 mM glutamine, 50 µg/ml uridine, vitamins, amino acids, and 1 mM sodium pyruvate in a humified, 5% $CO_2$ atmosphere at 37°C. iAstrocytes were differentiated from induced neural progenitor cells (iNPCs) as previously described (Meyer et al, 2014). iNPCs were cultured in DMEM containing 1% N2 supplement (Life Technologies), 1% B27 supplement, and 20 ng/ml fibroblast growth factor-2 (Preprotech). INPCs were differentiated into induced astrocytes (iAstrocytes) on 10 cm dishes coated with fibronectin (5 µg/ml, Millipore) by culturing in DMEM with 10% FBS and 0.3% N2. iNPCs were differentiated to iAstrocytes over 8 d.

### iPSC-derived motor neuron culture

iPSCs were cultured on vitronectin-coated plates in mTeSR-plus (Stem Cell Technologies). When they reached 90% confluence they were passaged 1:1 with Relesr (Stem Cell Technologies) onto a Matrigel-coated plate in mTeSR-plus supplemented with 10 µM Rho kinase inhibitor for 24 h. After 24 h, the media was replaced with basal media (50% neurobasal media, 50% KnockOut TM DMEM/F12, 0.5X N2, 0.5X B27, 1X Glutamax, 1% Penicillin streptomycin) supplemented with 3 µM CHIR, 2 µM DMH1 and 2 µM SB431542 for 6 d, with full media changes every 24 h. Media was then changed to basal media supplemented with 1 µM CHIR, 2 µM DMH1, 2 µM SB431542, 0.1 µM retinoic acid, and 0.5 µM purmorphamine for 5 d, with full media changes every 24 h. Cells were then passaged at a ratio of 1:12 onto Matrigel-coated plates in NPC expansion media (basal media supplemented with 3 µM CHIR, 2 µM DMH1, 2 µM SB431542, 0.1 µM retinoic acid, 0.5 µM purmorphamine, and 0.5 µM valproic acid) supplemented with 10 µM Rho kinase inhibitor for 24 h. Full media changes with NPC expansion media were then performed every other day. When the NPCs were 100% confluent, the media was changed to the motor neuron progenitor differentiation media (basal media supplemented with 0.5 µM retinoic acid and 0.1 µM purmorphamine) for 5 d, with media changes every other day. The motor neuron progenitors were then passaged and plated onto Matrigel-coated plates at a density of 130,000 cells/cm

2 in motor neuron differentiation media (basal media supplemented with 0.5 µM retinoic acid, 0.1 µM purmorphamine, 0.1 µM compound E, 10 ng/ml BDNF, 10 ng/ml CNTF, and 10 ng/ml IGF1) supplemented with 10 µM Rho kinase inhibitor for 24 h. Media was then changed every other day for 9 d. Cells were then used for experiments.

### Lentiviral production

Lentiviruses (LV) were propagated in HEK293T cells using the calcium phosphate method (Deglon et al, 2000). Viral titres were measured by qPCR. Genomic DNA isolated from transduced HeLa cells was used as a template for qPCR with Woodchuck Hepatitis Virus Posttranscriptional Regulatory Element (WPRE) primers to assess the number of copies of stably integrated lentiviruses. An LV carrying GFP of a known biological titre (FACS titration) was used as a reference.

### Lentiviral transduction

Primary cortical neurons from E16.5 C9-500 BAC transgenic mouse embryos were transduced with the indicated lentiviral vectors at a MOI of 10 on DIV4. Cells received a 50% media change at DIV7 and were maintained until DIV10. iPSC-derived motor neurons from C9ALS/FTD patients were transduced at day 28 of differentiation with an MOI of 20. Cells received a 50% media change every other day and were maintained for a further 5 d.

### SDS–PAGE and immunoblot

Unless otherwise stated, cells were washed once with PBS before lysing directly in ice-cold RIPA buffer (50 mM Tris–HCl pH 6.8, 150 mM NaCl, 1 mM EDTA, 1 mM EGTA, 2% [wt/vol] SDS, 0.5% [wt/vol] deoxycholic acid, 1% [wt/vol] Triton X-100, and protease inhibitor cocktail [Thermo Fisher Scientific]). Lysates were incubated on ice for 30 min before being clarified at 17,000$g$ for 20 min at 4°C. Protein concentrations were determined by BCA assay (Thermo Fisher Scientific).

Proteins were separated by SDS–PAGE using 4–20% gradient mini-PROTEAN TGX precast polyacrylamide gels (Bio-Rad). Gels were run at 150 V for ~1 h. Proteins were transferred to 0.2 µm nitrocellulose membranes (Whatmann) by electroblotting at 100 V for 30 min using the Bio-Rad Criterion blotter (Bio-Rad). After transfer, membranes were blocked for 1 h at RT in TBS with 5% fat-free milk powder (Sigma-Aldrich) and 0.1% Tween-20. Membranes were incubated with primary antibodies in blocking buffer for 1 h at RT or overnight at 4°C. Membranes were washed three times for 10 min in TBS with 0.1% Tween-20 before incubation with secondary antibodies in block buffer for 1 h at RT. Secondary antibodies used for immunoblotting were horseradish peroxidase (HRP)-coupled goat anti-rabbit and goat anti-mouse IgG (1:5,000; Dako, Agilent Technologies LDA), or Alexa Fluor 680 donkey anti-rabbit IgG and Alexa Fluor 790 donkey anti-mouse IgG (1:50,000; Jackson ImmunoResearch, Stratech Scientific Ltd.). After HRP-coupled antibody incubation, membranes were washed three times for 10 min in TBST and prepared for chemiluminescent signal detection with SuperSignal West Pico Chemiluminescent substrate (Thermo Fisher

Scientific) according to the manufacturer's instructions. Chemiluminescent signals were detected using an Odyssey Fc imaging system (LI-COR Biosciences). Fluorescent signals were detected using an Odyssey Fc imaging system. Signal intensities were quantified using ImageJ/Fiji or ImageStudio (LI-COR Biosciences).

### Dot-blot

For DPR expression analysis via dot-blot, cells were harvested directly into 2x concentrated Laemmli loading buffer and passed through a 25 G needle 20 times before boiling for 5 min at 95°C. Equal volumes of each sample were spotted to 0.2 μm nitrocellulose membrane presoaked in TBST using the 96-well Bio-Dot Microfiltration apparatus (Bio-Rad) under vacuum. Membranes were left to dry at RT before processing as a standard immunoblot as described.

### Immunoprecipitation

HeLa cells were washed once in PBS before being lysed in ice-cold BRB80 buffer (80 mM K-PIPES pH 6.8, 1 mM MgCl$_2$, 1 mM EDTA, 1% [wt/vol] NP-40, and protease inhibitor cocktail) for 1 h on a roller at 4°C. Lysates were cleared at 17,000$g$ for 20 min at 4°C and protein concentrations determined by BCA assay (Thermo Fisher Scientific). 2 mg total protein was incubated with 2 μg anti-FLAG antibodies in BRB80 buffer for 16 h at 4°C when rotating. Antibodies were captured by incubation with 10 μl of Protein G-Sepharose magnetic beads for 2 h at 4°C when rotating. Beads were washed five times in ice-cold BRB80 buffer before eluting proteins in 2x Laemmli buffer. Magnetic beads were isolated from solutions using the Extractman device (Gibson Scientific Ltd).

### Antibodies

Primary antibodies used were as follows:
Mouse anti-α-Tubulin (DM1A, WB: 1:10,000; Sigma-Aldrich)
Mouse anti-Flag (M2, WB: 1:2,000; Sigma-Aldrich)
Rabbit anti-GAPDH (14C10, WB: 1:2,000; Cell Signaling)
Mouse anti-GFP (JL8, WB: 1:5,000; Clontech)
Mouse anti-HA (HA-7, WB: 1:2,000; Sigma-Aldrich)
Rabbit anti-HA (H6908, IF: 1:1,000; Sigma-Aldrich)
Mouse anti-V5 (R960, WB: 1:5,000, IF: 1:1,000; Invitrogen)
Rabbit anti-RuvBL1 (A304-716A, WB: 1:2,000; Bethyl Laboratories)
Rabbit anti-RuvBL2 (A302-536A, WB: 1:1,000; Bethyl Laboratories)
Mouse anti-GFP (JL8, WB: 1:5,000; Clontech)

### *Drosophila* stocks and maintenance

*Drosophila* were raised on standard cornmeal–yeast–sucrose medium at 25°C on a 12-h light:dark cycle, unless otherwise stated. UAS-AP(1024)eGFP (Flybase ID FBti0213155), UAS-PR(1100)eGFP (Flybase ID FBti0213157), UAS-GA(1020)eGFP (Flybase ID FBti0213158), and UAS-GR(1136)eGFP (Flybase ID FBti0213156) were described previously (West et al, 2020). UAS-mCD8-GFP (RRID: BDSC_32184) and UAS-mKate2.CAAX (RRID: BDSC_55091) stocks were obtained from the Bloomington Drosophila Stock Center (BDSC). UAS-Pontin (F000819) and UAS-Reptin (F001385) stocks were obtained from the FlyORF Zurich ORFeome stock center (Bischof et al, 2013). GMR-Gal4 flies were a gift from Sean T. Sweeney (The University of York, UK). nSyb-gal4 flies were a gift from Chris Elliott (The University of York, UK). All experiments were performed using flies from at least three independent crosses, per genotype.

### Negative geotaxis assays

Negative geotaxis assays were performed as described previously (West et al, 2020; Bennett et al, 2023). Flies were placed, without anesthetization, inside glass boiling tubes within our custom apparatus with a white, backlit background. Flies were banged down to the bottom of the tubes to elicit the startle-induced negative geotaxis escape behaviour and videos recorded using a Logitech web cam using virtualDub software (30 frames per second, 30 s). Videos were analysed in ImageJ to determine the distance travelled over time.

### *Drosophila* activity monitors

*Drosophila* locomotor activity was measured using the TriKinetics *Drosophila* Activity Monitor 5M (DAM5M) system. Individual flies were placed in transparent glass tubes (~65 × 5 mm) containing ~5 mm of food at one end (5% sucrose, 2% agar) sealed at the food end with paraffin wax and a cotton wool stopper at the other end. Tubes were loaded in the DAM5M monitors following the manufacturer's instructions (www.trikinetics.com). Activity monitors were housed in an incubator maintained at 25°C on a 12-h light dark cycle. Flies were given ~12 h to acclimatise before recording activity for the subsequent period of 24 h. The DAMSystem3 acquisition software was used to record DAM5M monitor output using a 1 min read interval set to acquire "Moves" for each beam and tube. "Moves" are defined as when a fly enters one beam after exiting another. After 24 h of active recording flies were returned to standard fly food vials at 25°C on a 12-h light:dark cycle. Raw monitor files were scanned using the FileScan program (www.trikinetics.com) and total number of "moves" plotted for the 24-h active recording period. For sleep analysis, DAM5M monitor files were analysed using Rtivity (Silva et al, 2022). Sleep was classified as periods of inactivity 5 min or longer, as described previously (Shaw et al, 2000).

### Protein extraction from *Drosophila* heads

Flies pan-neuronally (nSyb-Gal4) co-expressing UAS-AP(1024)eGFP, UAS-PR(1100)eGFP, UAS-GA(1020)eGFP, UAS-GR(1136)eGFP, or UAS-mCD8-GFP with either UAS-mKate2.CAAX, UAS-Pontin, or UAS-Reptin were aged to 7 DPE, snap frozen on dry ice and heads removed via vortexing. Frozen heads were ground to powder using a cell-pestle and lysed in RIPA buffer containing 2% SDS (10 mM Tris–Cl [pH 8.0], 1 mM EDTA, 0.5 mM EGTA, 1% Triton X-100, 0.1% sodium deoxycholate, 2% SDS, 140 mM NaCl) for 20 min. Lysates were cleared via centrifugation and total protein quantified via BCA assays (Pierce BCA Protein Assay Kit; #23227; Thermo Fisher Scientific).

### MSD-ELISA

Poly(GA), Poly(GR), Poly(PR), Poly(PA), and Poly(GP) levels were determined by a Meso Scale Discovery technology sandwich ELISA using the MSD QUICKPLEX SQ120 platform (Meso Scale Technology). Purified rabbit polyclonal capture antibodies for Poly(GA), Poly(GR), Poly(PR), Poly(PA), and Poly(GP) from two rabbits were generated by custom synthesis from Eurogentec. The poly(DPR)x7-10 peptides used for custom antibody synthesis were serially diluted to generate a standard curve (0.125–40 ng/ml) in each assay plate. Optimal pairing was established using a second biotinylated Poly(GA), Poly(GR), Poly(PR), Poly(PA), or Poly(GP) detection antibody with sulfo-tag streptavidin substrate leading to generation of the electroluminescent signal read in the MesoScale Discovery instrument (following manufacturer's instructions). Each sample was prepared in ice-cold RIPA buffer (50 mM Tris–HCl pH 6.8, 150 mM NaCl, 1 mM EDTA, 1 mM EGTA, 2% [wt/vol] SDS, 0.5% [wt/vol] deoxycholic acid, 1% [wt/vol] Triton X-100, and protease inhibitor cocktail [Thermo Fisher Scientific]) and diluted to 1.5 mg/ml. Equal amounts of protein (100 ng for all Poly(GA), Poly(GR), Poly(PR), and Poly(GP) assays or 5 ng for Poly(PA) assays from *Drosophila* heads) were mixed with capture antibody coated on 96-well plates.

Briefly, multi-array plates were coated overnight with 30 $\mu$l capture antibody (2 $\mu$g/ml) in TBS at 4°C. Plates were then washed in TBS + 0.2% Tween-20 (TBST) and blocked for 2 h in 3% non-fat milk in TBST at RT shaking at 700 rpm (Illumina High-Speed Microplate Shaker), before being washed in TBST and incubated with calibrant or samples at 4°C overnight at 700 rpm. The plates were washed, incubated for 2 h at RT 700 rpm with 25 $\mu$l biotinylated detection antibody (2 $\mu$g/ml) and 0.5 ng/ml SULFO-TAG streptavidin R32AD-1 in blocking buffer. After the final wash, 150 $\mu$l 2x read buffer R92TD was added and then read. Raw absorbance readings (OD) were plotted directly or were normalised relative to the control.

### Statistical analysis

Calculations and statistical analysis were performed using Excel (Microsoft Corporation) and Prism 9 software (GraphPad Software Inc.). Details of statistical analysis can be found in the figure legends.

### Ethics statement

Ethical approval to use iPSCs or iNPCs is in place for this project (REC approval 12/YH/0330). All animal in vivo experiments described were performed according to the Animal (Scientific Procedures) Act 1986, under the Project Licenses Azzouz_40/3739 and P31C8CC9D. This experimental work involves studies on genetically modified viral vectors already approved by the Health and Safety Executive (Azzouz_GMO_2006-07).

## Supplementary Information

## Acknowledgements

This work was supported by the ARUK award (ARUK-PG2018B-005). This project has also received funding from the Innovative Medicines Initiative 2 JointUndertaking (JU) under grant agreement No 945473. The JU receives support from the European Union's Horizon 2020 research and innovation programme and EFPIA. M Azzouz is also supported by JPND-MRC (MR/V000470/1), the European Research Council grant (ERC Advanced Award no. 294745), MRC DPFS Award (129016), MRC/LifeArc award (MR/V030140/1), and LifeArc (No. 163978 and P2022-0004, Recipient: 176666) and CureAP4. GM Hautbergue acknowledges support from the Medical Research Council (MRC) research grant MR/W00416X/1, the Biotechnology and Biological Sciences Research Council (BBSRC) grant BB/S005277/1, and LifeArc over the course of this study. KJ De Vos was supported by the Alzheimer's Society project grant 260 (AS-PG-15-023) and an Alzheimer's Research UK Major project grant (ARUK-PG2019A-008). RJH West was supported by Alzheimer's Society grants (510 and 611).

### Author Contributions

CP Webster: conceptualization, data curation, formal analysis, supervision, investigation, visualization, methodology, project administration, and writing—original draft, review, and editing.
B Hall: data curation, formal analysis, and investigation.
OM Crossley: data curation, formal analysis, and investigation.
D Dauletalina: data curation.
M King: data curation and methodology.
Y-H Lin: resources and methodology.
LM Castelli: resources and methodology.
Z-L Yang: data curation and methodology.
I Coldicott: data curation and methodology.
E Kyrgiou-Balli: data curation and methodology.
A Higginbottom: data curation and methodology.
L Ferraiuolo: resources and methodology.
KJ De Vos: resources, methodology, and writing—review and editing.
GM Hautbergue: resources, methodology, and writing—review and editing.
PJ Shaw: resources and methodology.
RJH West: resources, data curation, formal analysis, investigation, methodology, and writing—original draft, review, and editing.
M Azzouz: conceptualization, resources, data curation, formal analysis, supervision, funding acquisition, investigation, methodology, project administration, and writing—original draft, review, and editing.

### Conflict of Interest Statement

CP Webster and M Azzouz are coinventors on patents filled in the USA (US20230038479) and Europe (EP4061933) for the use of gene therapy vectors containing RuvBL1 and/or RuvBL2 "...for the treatment of neurodegenerative diseases that result from expression of polymorphic repeat expansions of GGGGCC in the first intron of the *C9orf72* gene" (WO/2021/160464). GM Hautbergue, M Azzouz, and PJ Shaw are co-founders of Crucible Therapeutics Limited. M Azzouz is a co-founder of BlackfinBio Limited.

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
