## [Reviewer comments · Life Science Alliance]

Life Science Alliance

RuvBL1/2 reduce toxic dipeptide-repeat protein burden in multiple models of C9orf72-ALS/FTD

Christopher Webster, Bradley Hall, Olivia Crossley, Dana Dauletalina, Marianne King, Ya-Hui Lin, Lydia Castelli, Zih-Liang Yang, Ian Coldicott, Ergita Balli, Adrian Higginbottom, Laura Ferraiuolo, Kurt De Vos, Guillaume Hautbergue, Pamela Shaw, Ryan J. H. West, and Mimoun Azzouz

DOI: <https://doi.org/10.26508/lsa.202402757>

Corresponding author(s): Christopher Webster, University of Sheffield; Mimoun Azzouz, University of Sheffield; and Christopher Webster, University of Sheffield

Review Timeline:

Submission Date:	2024-04-05
Editorial Decision:	2024-05-20
Revision Received:	2024-10-08
Editorial Decision:	2024-11-05
Revision Received:	2024-11-22
Accepted:	2024-11-25

Transaction Report:

May 20, 2024

Re: Life Science Alliance manuscript #LSA-2024-02757-T

Dr. Christopher P. Webster
The University of Sheffield
Neuroscience
SITRAN
385A Glossop Road
Sheffield S10 2HQ
United Kingdom

Dear Dr. Webster,

Thank you for submitting your manuscript entitled "RuvBL1/2 reduce toxic dipeptide-repeat protein burden in multiple models of C9orf72- ALS/FTD" to Life Science Alliance. The manuscript was assessed by expert reviewers, whose comments are appended to this letter. We invite you to submit a revised manuscript addressing the Reviewer comments.

Thank you for this interesting contribution to Life Science Alliance. We are looking forward to receiving your revised manuscript.

Sincerely,

B. MANUSCRIPT ORGANIZATION AND FORMATTING:

Reviewer #1 (Comments to the Authors (Required)):

Christopher et al. demonstrated that decreasing RAN-DPR provides therapeutic benefits for C9orf72 ALS/FTD. In this project, RuvBL1 and RuvBL2 are implicated in the clearance of DPR aggregates. RuvBL1 and RuvBL2 reduce DPR in model cell lines, primary neurons from C9-500 mice, and patient iPSC models. Finally, overexpression of RuvBL2 effectively rescues the DPR-related motor phenotype in a *Drosophila* model. The results are well organised and address key topics; however, there are comments to resolve the ambiguity of the RuvBL1 and 2 for therapeutic effect on C9-ALS/FTD.

Comments.

RuvBL1/2 overexpression reduces DPR protein levels in vitro

1. The Author investigated the plasmids expressing AUG-driven synthetic, codon-optimized, V5-tagged 100 repeats of poly(GA), poly(GR) or poly(PR) DPRs into HeLa cells co-transfected with plasmids expressing FLAG-tagged RuvBL1 or HA-tagged RuvBL2.

- Schematic diagram of synthetic DPR might necessary for audience to understand the structure of the DPR and their corresponding tags.

- It is minor comment but the control DPR should be co transfected with reporter plasmid which tagged with Flag or HA to make a similar dual transfection condition in Figure 1.

2. Overexpression of FLAG-RuvBL1 and HA-RuvBL2 led to a significant reduction in the detectable level of poly(GA) and poly(GR) (Figure 1B and C), but had no effect on poly(PR) levels

- PR is not effective to the RuvBL. However, it is possible that this is due to the dose effect. It might be saturated. It will be ideal to test series of low dose for the dual transfection.

3. overexpression of RuvBL1 and RuvBL2 significantly reduced the detectable levels of V5-DPRs on dot-blot (Figure 1E), indicating RuvBL1/2 can impact the level of DPRs produced via RAN translation.

- Author should address clearly about which frame of DPR is tagged with V5 from G4C2x45 in Figure 1E.

RuvBL1/2 are differentially expressed in C9orf72 patient cells

1. RuvBL1 protein was found to be significantly lower in all C9orf72-ALS derived patient cells compared to their controls (Figure 2A, B and C), while RuvBL2 levels were significantly reduced in two out of the three patient lines

2. A similar pattern was observed in the results of the RT-qPCR analysis which indicated a reduced level of expression of RuvBL1 mRNAs across all patient lines, and a reduced levels of RuvBL2 expression in one of the three patient lines (Figure 2D, E and F).

- It might necessary to address the C9-ALS patient tissues. Which part of the tissues and post-mortem stage and age as well. 3 patient might not enough, recommend to use 10 patients to show the significant differences.

Regarding the FRqPCR, this the RvBL1 and 2 normalised with GAPDH? What was the normalisation methods?

RuvBL1/2 overexpression reduces DPRs in primary cortical neurons from C9-500 BAC mice

1. overexpression of RuvBL1 or RuvBL2 via lentiviral (LV) transduction could reduce poly(GA) and poly(GP) levels in these cells. C9-500 BAC primary cortical neurons were transduced with LV-GFP, LV-FLAG-RuvBL1 or LV-HA-RuvBL2 at DIV4, before proteins were harvested at DIV10 and LV transduction confirmed by immunoblot (Figure 3A, B and C)

- Good model

2. Levels of poly(GA) and poly(GP) were measured by MSD-ELISA (Figure 3D and E). As expected there was a significant detection of poly(GA) and poly(GP) signals in the C9-500 BAC neurons compared to the wild-type controls. After transduction with LV-FLAG-RuvBL1 and LV-HA-RuvBL2, we demonstrated that RuvBL1 and RuvBL2 overexpression significantly reduced poly(GA) DPRs (Figure 3D)

- It is great to see the level of the DPR in cell lysate but the MSD measure soluble form of DPR. The RuvBL decrease the aggregation so the dot blot is more ideal assay for the study.

3. only transduction with HA-RuvBL2 was able to significantly reduced poly(GP) levels in these assays (Figure 4E).
- Dot blot will make this finding stronger.

RuvBL1/2 overexpression reduces poly(GA) DPRs in patient iPSC derived motor neurons

1. Overexpression of RuvBL2 led to a significant reduction in the levels of detectable poly(GA) DPRs compared to the GFP transduced control cells (Figure 4C). However, RuvBL2 had no significant impact on poly(GP) levels in these assays (Figure 4D)
- Regarding the iPSC, expression of the DPR is depend on how well differentiated to the neuron and how long incubated. It will be ideal to add schematic diagram of time line of the iPSC work for the assay.

RuvBL1/2 overexpression reduces DPR proteins in a Drosophila model of C9ALS/FTD

1. 7 days post eclosion (DPE) proteins were extracted from fly heads, and levels of each DPR measured by MSD-ELISA to accurately assess changes in DPR levels between groups. Co-expression of RuvBL1 with GA and GR had no effect on detectable DPR levels (Figure 5A and B)

- I am not expert for the fly, so not able to comment.

2. was able to significantly reduce poly(PR) and poly(PA) levels (Figure 5C and D).

-I am not expert for the fly, so not able to comment.

RuvBL2 co-expression rescues GR(1000), PR(1000) and PA(1000) associated motor phenotypes in Drosophila

1. pan-neuronal expression of PA(1000) and GR(1000) led to a significant decrease in vertical climbing distance from 7 to 14 DPE in the mKate co- expressing groups (Figure 6B and C)

2. PR expressing flies did not exhibit a progressive reduction in climbing between 7 and 14 DPE, co-expression of RuvBL2 in these flies did lead to a significant increase in climbing distance at both 7 and 14 DPE (Figure 6D).

3. Flies co-expressing mKate with PA(1000), GR(1000) or PR(1000) displayed a significant reduction in activity compared to the GFP control (Figure 6E and F)

4. Using the Rtivity software to measure total time sleeping over the 24 h period we discovered that both GR(1000) and PR(1000) flies sleep more during daylight hours (Figure 6G)

5. Co-expression of Reptin was able to rescue this defect in PR(1000) flies and partially rescue in the GR(1000) flies (Figure 6G)

- It will be excellent to draw a schematic diagram for the study to understand the procedure for the study.

RuvBL1/2 overexpression slows the rate of DPR production by affecting transcription

1. HeLa cells transfected 24 h previously with control, FLAG-RuvBL1 or HA-RuvBL2 plasmids, were transfected with V5-sense DPR plasmids and protein translation inhibited with CHX 8 h post transfection. RuvBL1 and RuvBL2 overexpression was confirmed by immunoblot (Figure 7A)

- There is no V5 WB blot to show the DPR.

2. Cyclin D1, having a short half-life due to rapid turnover, was used as an indicator of translational inhibition and protein clearance (Figure 7A)

3. In these assays we again determined poly(GP) levels via MSD-ELISA to give the most accurate measure of total DPR proteins (Figure 7B)

-Howe about the GA and GR? Here is showing only GP.

4. We discovered that after translational inhibition with CHX the total level of poly(GP) DPR remained stable, and showed no significant level of clearance over the 24 h period studied (Figure 7B)

- Might necessary to test the GA and GR.

5. Sense plasmids were delivered to HeLa cells which had been transfected 24 h previously with control, FLAG-RuvBL1 or HA-RuvBL2 plasmids. Proteins were then harvested at a range of time points over the next 24 h and poly(GP) levels determined via MSD-ELISA. RuvBL1/2 protein overexpression was confirmed by immunoblot (Figure 7C)

- V5 western blot or dot blot might necessary to compare with MSD.

6. The levels of poly(GP) at each time point are shown in Figure 7D. The presence of RuvBL1 and RuvBL2 appeared to slow the rate of DPR production and, indeed, 24 hours post DPR transfection there was significantly less poly(GP) in RuvBL2 overexpressing cells compared to control (Figure 7E).

-

7. HeLa cells were transfected with control plasmid or FLAG-tagged RPL10a, before isolating RPL10a with anti-FLAG antibodies and probing the resulting immunoprecipitate for RuvBL1 and RuvBL2. Endogenous RuvBL1, and to a greater extent, RuvBL2 were found to specifically co-immunoprecipitate with RPL10a (Figure 7F and G), indicating RuvBL1/2 were able to interact with the translational machinery.

- What is the rationale for the RPL10 IP? If the RPL10a is involved in RuvB1,2 mechanism for the DPR aggregation, author should use endogenous lysate. The overexpression might increase the chance of false positive result.

Figure 8

1. we discovered overexpression of RuvBL1 and RuvBL2 had a profound effect on sense DPR transcription (Figure 8B).
 - Is RuvBL1 and 2 decrease the mRNA of C9RAN? Have you measured the intronic sequence?
 - What is the rationale for this ?
2. This effect did not appear to be a reduction in global transcription as GAPDH expression was unaffected (Figure 8C)
 - Great to test .
3. Furthermore, the expression of C9orf72 was also unaffected by RuvBL1 and RuvBL2 overexpression (Figure 8D).
 - Great to test the coding gene of C9orf72 mRNA.
4. these data indicated that overexpression of RuvBL1 and RuvBL2 are able to reduce transcription of the C9orf72 repeat, leading to reduced DPR translation

It is not clear what exactly is measured in Figure 8. It would be excellent to add a schematic diagram illustrating what exactly is addressed for the C9 mRNA.

Reviewer #2 (Comments to the Authors (Required)):

Major criticism

In their manuscript, Webster et al claims RuvBL1/2 reduce toxic dipeptide-repeat protein burden in multiple models of C9orf72-ALS/FTD. Overall, the reviewer recognizes this study incomplete and requires extensive reconfiguration before publication. While the reduced endogenous expressions of AAA+ family members RuvBL1/2 in C9ALS/FTD fibroblast-derived iAstrocyte is interesting, the overexpression of RuvBL1/2 in iPSC-derived motoneurons had only marginal impact, if any, on DPR expression and this result obscures the importance of RuvBL1/2 for DPR expression in endogenous context. Although multiple models were used (HeLa cells, primary cultured neuron from C9 model mouse, and C9 model Drosophila), most experiments relied solely on RuvBL1/2 overexpression system. Counter experiments examining the effects of reduced RuvBL1/2 expression on DPR expressions are essential for at least for some key experiments. Unfortunately, the species of DPRs that showed an effect on RuvBL1/2 overexpression was not consistent from one experimental system to another, and the possible reasons for the discrepancies were not satisfactorily explained. Most importantly, they clearly showed CMV promoter-based RuvBL1/2 overexpression prominently suppressed CMV promoter-derived repeat RNA (Fig 8B). The experiments in Fig. 1E and Fig. 8 are paired experiments, suggesting that the reduction in repeat RNA levels is behind the suppression of DPR expression observed in Fig. 1E. The endogenous GAPDH and/or C9orf72 mRNA levels are unsuitable as controls. It needs to be clarified whether the observed effect is sequence specific for GGGGCC repeat containing transcript or a general phenomenon in this promoter context (e.g. by using GFP sequence instead of the repeat sequence). In addition, the effects of repeat RNA expression levels on RuvBL1/2 overexpression/reduction must be experimentally verified in all models used in order to correctly interpret their results. Lastly, the introduction/discussion section focused on protein disaggregation/clearance properties of RuvBL1/2. However, their results clearly show most prominent effect of RuvBL1/2 overexpression is on RNA expression levels (transcription/metabolism including NMD); the authors should introduce/discuss more details about the effect of RuvBL1/2 at the RNA level.

Minor points

The manuscript contains inaccurate generalizations (oversimplification) in various parts that overexpression of RuvBL1/2 reduces DPR even when the data do not support this.

p8. last line. 2 μ DMH1 : Correction needed.

p17 line 14-16 "Similar to what was seen with LV-FLAG-RuvBL1 transduction in primary cortical neurons, overexpression of RuvBL1 had no impact on poly(GA) or poly(GP) levels in these experiments." This statement is incorrect. Fig 3D shows a significant reduction of GA DPR upon RuvBL1 overexpression.

p17-18 and Figure 5: Expressions of the gene names of Drosophila orthologs are inconsistent. In the text, it is described as RuvBL2, but in Fig. it is marked as Reptin.

p18 line 2 Inconsistency of expression p18 line 2 "poly(PA)" Fig 5D "AP"

p20-21 In the experiments in Figure 7F, G (RPL10A and RuvB1/2 co-IP experiments), there is no evidence that the overexpressed RPL10A is actually incorporated into the translational ribosome. So the reviewer thinks the authors cannot claim that "RuvBL1/2 were able to interact with the translational machinery" based solely on this experiment.

Figure 1: Endogenous expression levels of RuvBL1/2 need to be shown.

Figure 2 D, E, F and Figure 8 B, C, D lack each data point

Figure 2 legend (D-F). Is one-way ANOVA with Tukey post-test correct ?

Figure 3A-C untreated: untreated? Correction needed.

Figure 6C Why the numbers (data points) of GR1000 mKate flies are small?

Figure 7A cyclin D blot signals are too weak.

Reviewer #1 (Comments to the Authors (Required)):

Christopher et al. demonstrated that decreasing RAN-DPR provides therapeutic benefits for C9orf72 ALS/FTD. In this project, RuvBL1 and RuvBL2 are implicated in the clearance of DPR aggregates. RuvBL1 and RuvBL2 reduce DPR in model cell lines, primary neurons from C9-500 mice, and patient iPSC models. Finally, overexpression of RuvBL2 effectively rescues the DPR-related motor phenotype in a *Drosophila* model. The results are well organised and address key topics; however, there are comments to resolve the ambiguity of the RuvBL1 and 2 for therapeutic effect on C9-ALS/FTD.

We thank Reviewer #1 for their kind and comprehensive review of our manuscript. We also very much appreciate their positive assessment and encouraging feedback. We have addressed their comments point by point below.

Comments.

RuvBL1/2 overexpression reduces DPR protein levels in vitro 1. The Author investigated the plasmids expressing AUG-driven synthetic, codon-optimized, V5-tagged 100 repeats of poly(GA), poly(GR) or poly(PR) DPRs into HeLa cells co-transfected with plasmids expressing FLAG-tagged RuvBL1 or HA-tagged RuvBL2.

- Schematic diagram of synthetic DPR might necessary for audience to understand the structure of the DPR and their corresponding tags.

We thank Reviewer 1 for this comment. We agree that a schematic diagram would be helpful in understanding the structure of these synthetic DPRs. These constructs have all been published and described previously¹⁻³. We have now included a schematic diagram as Supplementary Figure 1 to demonstrate the structure of the synthetic DPRs, referred to on Page 15 Line 409 of the revised manuscript. We also include a schematic showing the structure of the RAN-translated pure repeat DPRs in Supplementary Figure 3, referred to on Page 15 Line 425. This figure is adapted from our previously published schematic in Bauer et al. 2022 and Castelli et al. 2023^{1,2}.

- It is minor comment but the control DPR should be co transfected with reporter plasmid which tagged with Flag or HA to make a similar dual transfection condition in Figure 1.

In these experiments we chose to co-transfect an empty pcDNA3.1 plasmid which had the same plasmid backbone, including CMV promoter, as the FLAG-RuvBL1 and HA-RuvBL2 expression plasmids. In doing so the same total DNA concentration was delivered to the cells. In our hands, transfection of control plasmids containing a tag only does not lead to detectable expression of said tag. We therefore chose to use the empty vector only as control.

2. Overexpression of FLAG-RuvBL1 and HA-RuvBL2 led to a significant reduction in the detectable level of poly(GA) and poly(GR) (Figure 1B and C), but had no effect on poly(PR) levels

- PR is not effective to the RuvBL. However, it is possible that this is due to the dose effect. It might be saturated. It will be ideal to test series of low dose for the dual transfection.

We thank Reviewer 1 for this interesting suggestion. We have now performed repeat experiments where the total amount of transfected V5-PR100 plasmid was titrated down while keeping the level of RuvBL1 or RuvBL2 overexpression constant (Supplementary Figure 2). In these assays we did not observe an effect of RuvBL1/2 overexpression on V5-PR100 levels even at lower V5-PR100 “doses”. These data indicate that the inability of RuvBL1/2 to affect PR100 levels was not due to a saturation effect of this DPR. These data are included in Supplementary Figure 2A and are referred to in the revised manuscript on Page 15 Line 422.

3. overexpression of RuvBL1 and RuvBL2 significantly reduced the detectable levels of V5-DPRs on dot-blot (Figure 1E), indicating RuvBL1/2 can impact the level of DPRs produced via RAN translation.

- Author should address clearly about which frame of DPR is tagged with V5 from G4C2x45 in Figure 1E.

We thank Reviewer 1 for the opportunity to resolve this confusion. As described on Page 15 Line 424 the V5-tag is located in all 3 reading frames downstream of the 45xG4C2 repeat. Further to this, our schematic diagram included in Supplementary Figure 3 (as mentioned above) now also demonstrates the structure of this RAN translated construct.

RuvBL1/2 are differentially expressed in C9orf72 patient cells

1. RuvBL1 protein was found to be significantly lower in all C9orf72-ALS derived patient cells compared to their controls (Figure 2A, B and C), while RuvBL2 levels were significantly reduced in two out of the three patient lines.
2. A similar pattern was observed in the results of the RT-qPCR analysis which indicated a reduced level of expression of RuvBL1 mRNAs across all patient lines, and a reduced levels of RuvBL2 expression in one of the three patient lines (Figure 2D, E and F).

- It might necessary to address the C9-ALS patient tissues. Which part of the tissues and post-mortem stage and age as well. 3 patient might not enough, recommend to use 10 patients to show the significant differences.

We apologise if these experiments were not clear. The patient cells used in Figure 2 were induced Neural Progenitor Cells (NPCs), reprogrammed from patient fibroblast cells and differentiated into iAstrocyte cells. These were not patient brain tissues, and so post-mortem stage and age are not applicable to these data. We are unable to increase the number of patient lines used, but 3 controls and 3 C9orf72-ALS/FTD patient lines has been is standard methodology in our other publications^{1,4,5}. In reviewing this data, it was discovered that incorrect qPCR primer pairs were used in this analysis. We therefore repeated the RT-qPCR analysis of RuvBL1/2 levels in these patient lines. These updated qPCR data replace the original qPCR data in Figure 2. As requested by Reviewer 2 below these graphs now show the individual data points.

Regarding the FRqPCR, this the RvBL1 and 2 normalised with GAPDH? What was the normalisation methods?

In these RT qPCR experiments 18S rRNA was used as the reference gene. Levels of RuvBL1 or RuvBL2 mRNA were quantified relative to 18S according to the delta-delta-Ct method. Expression levels are shown relative to each control line.

RuvBL1/2 overexpression reduces DPRs in primary cortical neurons from C9-500 BAC mice

1. overexpression of RuvBL1 or RuvBL2 via lentiviral (LV) transduction could reduce poly(GA) and poly(GP) levels in these cells. C9-500 BAC primary cortical neurons were transduced with LV-GFP, LV-FLAG-RuvBL1 or LV-HA-RuvBL2 at DIV4, before proteins were harvested at DIV10 and LV transduction confirmed by immunoblot

(Figure 3A, B and C)

- Good model

We thank Reviewer 1 for the encouraging words.

2. Levels of poly(GA) and poly(GP) were measured by MSD-ELISA (Figure 3D and E). As expected there was a significant detection of poly(GA) and poly(GP) signals in the C9-500 BAC neurons compared to the wild-type controls. After transduction with LV-FLAG-RuvBL1 and LV-HA-RuvBL2, we demonstrated that RuvBL1 and RuvBL2 overexpression significantly reduced poly(GA) DPRs (Figure 3D)

- It is great to see the level of the DPR in cell lysate but the MSD measure soluble form of DPR. The RuvBL decrease the aggregation so the dot blot is more ideal assay for the study.

We thank Reviewer 1 for their insightful comment. However, as shown in the literature, MSD-ELISA has also been utilised for the detection of both soluble and insoluble C9orf72-associated DPR proteins^{6,7}, provided an appropriate solubilising buffer is used to extract the insoluble DPRs. In this study we utilised RIPA lysis buffer supplemented with 2% SDS, a concentration 20x higher than that typically found in standard RIPA buffer. Previous studies have demonstrated that 2% SDS is sufficient to solubilise insoluble protein aggregates rendering them suitable ligands for ELISA⁸. In our hands we find that our method of lysis in 2% SDS-RIPA, followed by sonication efficiently solubilises all C9orf72-associated DPR proteins. Furthermore, clarification of these lysates via centrifugation does not yield a pellet corresponding to an insoluble fraction. Finally, MDS-ELISA has become the gold standard method for DPR detection in the study of C9orf72 pathogenesis and is routinely used over all other available methods. Given that dot blots are semi-quantitative, at best, we chose to utilise MSD-ELISA for the majority of our assays, particularly where small differences were likely, or where endogenous or RAN-translated DPR levels were being detected.

3. only transduction with HA-RuvBL2 was able to significantly reduced poly(GP) levels in these assays (Figure 4E).

- Dot blot will make this finding stronger.

As described above, our method of cell lysis leads to efficient recovery of soluble and insoluble DPR proteins and allows us to detect small differences in DPR levels which a semi-quantitative dot-blot would not. Furthermore, as described via Salomonsson et

al., caution should be taken with the use of commercially available antibodies for endogenous DPR detection as many have been generated and validated through positive selection of recombinant protein⁹, and therefore may not be appropriate for detection of endogenously produced DPRs. In our hands we are unable to detect endogenous DPRs using commercially available antibodies via dot blot. However, given the sensitivity of our MSD-ELISA we do not view this as an issue.

RuvBL1/2 overexpression reduces poly(GA) DPRs in patient iPSC derived motor neurons

1. Overexpression of RuvBL2 led to a significant reduction in the levels of detectable poly(GA) DPRs compared to the GFP transduced control cells (Figure 4C). However, RuvBL2 had no significant impact on poly(GP) levels in these assays (Figure 4D) - Regarding the iPSC, expression of the DPR is depend on how well differentiated to the neuron and how long incubated. It will be ideal to add schematic diagram of time line of the iPSC work for the assay.

We thank Reviewer 1 for this suggestion. We have now included a time-line of these experiments in revised Figure 4A, supporting the methodology description provided on pages 8 and 9 relating to iPSC neuronal culture and transduction.

RuvBL1/2 overexpression reduces DPR proteins in a Drosophila model of C9ALS/FTD

1. 7 days post eclosion (DPE) proteins were extracted from fly heads, and levels of each DPR measured by MSD-ELISA to accurately assess changes in DPR levels between groups. Co-expression of RuvBL1 with GA and GR had no effect on detectable DPR levels (Figure 5A and B)

- I am not expert for the fly, so not able to comment.

2. was able to significantly reduce poly(PR) and poly(PA) levels (Figure 5C and D).

-I am not expert for the fly, so not able to comment.

RuvBL2 co-expression rescues GR(1000), PR(1000) and PA(1000) associated motor phenotypes in Drosophila

1. pan-neuronal expression of PA(1000) and GR(1000) led to a significant decrease in vertical climbing distance from 7 to 14 DPE in the mKate co- expressing groups (Figure 6B and C)

2. PR expressing flies did not exhibit a progressive reduction in climbing between 7 and 14 DPE, co-expression of RuvBL2 in these flies did lead to a significant increase in climbing distance at both 7 and 14 DPE (Figure 6D).

3. Flies co-expressing mKate with PA(1000), GR(1000) or PR(1000) displayed a significant reduction in activity compared to the GFP control (Figure 6E and F)

4. Using the Rtivity software to measure total time sleeping over the 24 h period we discovered that both GR(1000) and PR(1000) flies sleep more during daylight hours (Figure 6G)

5. Co-expression of Reptin was able to rescue this defect in PR(1000) flies and partially rescue in the GR(1000) flies (Figure 6G)

- It will be excellent to draw a schematic diagram for the study to understand the procedure for the study.

We thank Reviewer 1 for their suggestion. We have now included a timeline in Supplementary Figure 8 detailing when each assay was performed post-eclosion. Supplementary Tables 1 and 2 also indicate the *Drosophila* genotypes used in this study as well as the genetic crosses used to generate the different flies for each figure.

RuvBL1/2 overexpression slows the rate of DPR production by affecting transcription

1. HeLa cells transfected 24 h previously with control, FLAG-RuvBL1 or HA-RuvBL2 plasmids, were transfected with V5-sense DPR plasmids and protein translation inhibited with CHX 8 h post transfection. RuvBL1 and RuvBL2 overexpression was confirmed by immunoblot (Figure 7A)

- There is no V5 WB blot to show the DPR.

Given the aggregating nature of these DPRs, reliable western blots for V5 detection were challenging. As we were investigating small changes in DPR levels in these assays, we chose to utilise MSD-ELISA analysis of DPRs in these experiments. The same lysates used for western blot were also used for the MSD-ELISA analysis, therefore the presence of the DPRs is demonstrated in Figures 7B and D.

2. Cyclin D1, having a short half-life due to rapid turnover, was used as an indicator of translational inhibition and protein clearance (Figure 7A)

As pointed out by Reviewer 2 below the cyclin D level appeared very weak in this figure. We have now amended this figure after re-running the samples to obtain better

visualisation of the cyclin D levels.

3. In these assays we again determined poly(GP) levels via MSD-ELISA to give the most accurate measure of total DPR proteins (Figure 7B)

-Howe about the GA and GR? Here is showing only GP.

We have previously demonstrated all three sense DPR species (poly-GA, GR and GP) are produced from this V5-45xG4C2 RAN construct^{1,2}, with poly-GP levels being the most abundant. Given that we were investigating relatively small differences in the levels of detectable DPRs we therefore focussed our analysis on the levels of poly-GP only. In this assay poly GP was used as an indicator of total RAN translated DPRs and a proxy for poly-GA and poly-GR levels.

4. We discovered that after translational inhibition with CHX the total level of poly(GP) DPR remained stable, and showed no significant level of clearance over the 24 h period studied (Figure 7B)

- Might necessary to test the GA and GR.

As described above we chose to focus on the levels of RAN translated poly-GP DPRs in these assays as these are more abundant than poly-GA and GR making detection more reliable.

5. Sense plasmids were delivered to HeLa cells which had been transfected 24 h previously with control, FLAG-RuvBL1 or HA-RuvBL2 plasmids. Proteins were then harvested at a range of time points over the next 24 h and poly(GP) levels determined via MSD-ELISA. RuvBL1/2 protein overexpression was confirmed by immunoblot (Figure 7C)

- V5 western blot or dot blot might necessary to compare with MSD.

Given the semi quantitative nature of dot blots, the solubilising nature of our lysis buffer, and the sensitivity of our MSD-ELISA assay, quantification of DPR levels by our in-house MSD-ELISA was far superior to western or dot blot detection. In these experiments, levels of DPR were determined at the indicated times post transfection. Given that these cells had only been transfected for a matter of hours the levels of DPR that had been produced via RAN translation in that time period were extremely low, necessitating their detection via a more sensitive assay than western or dot-blot, namely MSD ELISA. Furthermore, in our hands many of the commercially available

DPR antibodies do not reliably detect DPRs at the level produced via RAN translation.

6. The levels of poly(GP) at each time point are shown in Figure 7D. The presence of RuvBL1 and RuvBL2 appeared to slow the rate of DPR production and, indeed, 24 hours post DPR transfection there was significantly less poly(GP) in RuvBL2 overexpressing cells compared to control (Figure 7E).

-

7. HeLa cells were transfected with control plasmid or FLAG-tagged RPL10a, before isolating RPL10a with anti-FLAG antibodies and probing the resulting immunoprecipitate for RuvBL1 and RuvBL2. Endogenous RuvBL1, and to a greater extent, RuvBL2 were found to specifically co-immunoprecipitate with RPL10a (Figure 7F and G), indicating RuvBL1/2 were able to interact with the translational machinery.

- What is the rationale for the RPL10 IP? If the RPL10a is involved in RuvBL1,2 mechanism for the DPR aggregation, author should use endogenous lysate. The overexpression might increase the chance of false positive result. We thank Reviewer 1 for their comment and appreciate the opportunity to resolve this issue. The data presented in Figure 7 indicated that RuvBL1/2 overexpression was slowing the rate of DPR production, rather than increasing the rate of DPR clearance. An effect on DPR production could be due to an effect on translation, or due to an effect on transcription. We initially explored a potential link between RuvBL1/2 and protein translation by investigating whether RuvBL1/2 were able to interact with the translational machinery. In these immunoprecipitation assays we found that endogenous RuvBL1/2 co-immunoprecipitated with overexpressed RPL10a, a member of the 60S ribosomal subunit. These data suggested a possible interaction between RuvBL1/2 and the translational machinery. We agree with Reviewer 1 that overexpression might increase the chance of a false positive result. However, these data preceded the data obtained in Figure 8, which strongly indicated that RuvBL1/2 were mainly impacting transcriptional regulation, hence an overall decrease in DPR production. These findings lessened the importance of the findings of Figure 7F and G. Accordingly, and in agreement with suggestions from Reviewer 2 (see below), we have toned down the language used to describe this potential interaction (Page 22 Paragraph beginning on Line 623), and emphasised the link to transcriptional

regulation and nonsense mediated decay (NMD) in our discussion (Pages 26 and 27).

Figure 8

1. we discovered overexpression of RuvBL1 and RuvBL2 had a profound effect on sense DPR transcription (Figure 8B).

- Is RuvBL1 and 2 decrease the mRNA of C9RAN? Have you measured the intronic sequence?

- What is the rationale for this ?

Given the published roles of RuvBL1/2 in transcriptional regulation, we investigated the level of 45xG4C2 RAN DPR producing mRNA transcripts via qPCR, utilising primers that bind immediately downstream of the G4C2 repeat. Quantifying the levels of these transcripts indicated that overexpression of RuvBL1/2 significantly reduced their levels. In line with Reviewer 2's comments below we repeated these assays with an EGFP control expressing vector to ensure this was not an effect of promoter usage (Supplementary Figure 9). RuvBL1/2 had no effect on EGFP expression, while Sense DPR transcripts were again significantly reduced. Because RuvBL1/2 did not appear to be affecting transcription from CMV promoters in general, we hypothesise that their effect on C9 RAN transcript levels is likely via another mechanism potentially nonsense mediated decay and mRNA metabolism. As suggested by Reviewer 2 below, we have now included more on this in our discussion in the revised manuscript (Pages 26 and 27).

2. This effect did not appear to be a reduction in global transcription as GAPDH expression was unaffected (Figure 8C)

- Great to test .

3. Furthermore, the expression of C9orf72 was also unaffected by RuvBL1 and RuvBL2 overexpression (Figure 8D).

- Great to test the coding gene of C9orf72 mRNA.

4. these data indicated that overexpression of RuvBL1 and RuvBL2 are able to reduce transcription of the C9orf72 repeat, leading to reduced DPR translation. It is not clear what exactly is measured in Figure 8. It would be excellent to add a schematic diagram illustrating what exactly is addressed for the C9 mRNA.

Apologies if this figure was unclear. Figure 8 (and now Supplementary Figures 9 and 10) focus on qPCR analysis of the indicated transcripts. Overexpression of RuvBL1/2 significantly reduced 45xG4C2 RAN DPR producing transcripts, without affecting endogenous C9orf72 or GAPDH transcripts, or exogenous EGFP transcripts. The location of the primers used to measure these 45xG4C2 containing transcripts is now indicated in the schematic included in Supplementary Figure 9A.

Reviewer #2 (Comments to the Authors (Required)):

Major criticism:

In their manuscript, Webster et al claims RuvBL1/2 reduce toxic dipeptide-repeat protein burden in multiple models of C9orf72-ALS/FTD. Overall, the reviewer recognizes this study incomplete and requires extensive reconfiguration before publication. While the reduced endogenous expressions of AAA+ family members RuvBL1/2 in C9ALS/FTD fibroblast-derived iAstrocyte is interesting, the overexpression of RubBL1/2 in iPSC-derived motoneurons had only marginal impact, if any, on DPR expression and this result obscures the importance of RuvBL1/2 for DPR expression in endogenous context.

We thank Reviewer 2 for taking the time to give such a thorough analysis and appraisal of our manuscript. We have endeavoured to address all points raised and hope that the extensive additions we have made to our manuscript will alleviate the concerns of Reviewer 2.

Although multiple models were used (HeLa cells, primary cultured neuron from C9 model mouse, and C9 model Drosophila), most experiments relied solely on RuvBL1/2 overexpression system. Counter experiments examining the effects of reduced RuvBL1/2 expression on DPR expressions are essential for at least for some key experiments.

We thank Reviewer 2 for this key point. Consequently, we obtained commercially available siRNAs targeting endogenous RuvBL1 and RuvBL2 from Dharmacon. After knocking down RuvBL1 or RuvBL2 expression in HeLa cells we delivered our V5-45xG4C2 RAN producing DPR construct via transfection. We then measured poly-GP DPR levels by MSD-ELISA 24 hours post transfection. Knockdown of RuvBL1 and RuvBL2 was confirmed by western blot. These new data form Supplementary Figure 6 and are described in detail on Pages 16 and 17 beginning at Line 458 of our revised manuscript. Knockdown of RuvBL1 or RuvBL2 did not affect the level of poly-GP DPRs measured by MSD-ELISA.

We also performed siRNA-based experiments in relation to the transcriptional regulation of the V5-45xG4C2 repeat (Supplementary Figure 10). RuvBL1 targeted siRNA resulted in a significant reduction in RuvBL1 and RuvBL2 mRNA levels, while RuvBL2 siRNA significantly reduced RuvBL2 mRNA levels only. In both cases there

was no effect on the level of transcription from the V5-45xG4C2 repeat. This is described on Page 23 Line 659).

Unfortunately, the species of DPRs that showed an effect on RuvBL1/2 overexpression was not consistent from one experimental system to another, and the possible reasons for the discrepancies were not satisfactorily explained.

We thank Reviewer 2 for their comment. To address this, we have now included greater justification of why different DPRs were assessed in the text (Page 17 Line 486 and Page 18 Line 514). In brief, whenever DPRs formed by RAN translation, either after transfection or at the endogenous level, were being assessed we chose to determine poly(GP) levels. Poly(GP) DPRs are the most abundant DPR produced by RAN translation of the C9orf72 repeat expansion, but given that RAN translation is inefficient, detectable levels can still be extremely low. We therefore focussed on this species to give the most reliable detection of DPR levels overall. In the C9-500 BAC cortical neurons and patient iPSC derived motor neurons, we chose to focus on poly(GA) as well as poly(GP). We have found that the levels of other DPRs in these cells are very low, making reliable detection, even via MSD-ELISA, problematic. In the case of our *Drosophila* work we made use of the four DPR producing *Drosophila* lines generated by West et al. 2020¹⁰. These *Drosophila* lines overexpress the different DPRs and so they are produced at levels that can be reliably detected by the MSD-ELISAs we have available. There is currently only one *Drosophila* poly(GP) DPR model which expresses shorter DPR repeats and is not publicly available¹¹.

Most importantly, they clearly showed CMV promoter-based RuvBL1/2 overexpression prominently suppressed CMV promoter-derived repeat RNA (Fig 8B). The experiments in Fig. 1E and Fig. 8 are paired experiments, suggesting that the reduction in repeat RNA levels is behind the suppression of DPR expression observed in Fig. 1E. The endogenous GAPDH and/or C9orf72 mRNA levels are unsuitable as controls. It needs to be clarified whether the observed effect is sequence specific for GGGGCC repeat containing transcript or a general phenomenon in this promoter context (e.g. by using GFP sequence instead of the repeat sequence).

We would like to thank Reviewer 2 for highlighting this important control. Accordingly, we repeated RuvBL1/2 overexpression experiments and co-transfected EGFP-C2 plasmid as an alternative negative control to the empty vector only. Most importantly

for these experiments the EGFP was also under the control of a CMV promoter. RuvBL1/2 overexpression had no effect of EGFP protein levels as shown in Supplementary Figure 2B and referred to on Page 16 Line 433. When we repeated these experiments to look at EGFP transcriptional regulation we also did not observe any effect of RuvBL1/2 overexpression, despite the fact that C9-RAN mRNA levels were again significantly reduced. These new data form Supplementary Figure 9 and are described on Page 23 beginning on Line 648 of the revised manuscript.

In addition, the effects of repeat RNA expression levels on RuvBL1/2 overexpression/reduction must be experimentally verified in all models used in order to correctly interpret their results.

We thank Reviewer 2 for this comment and agree that understanding how the presence of the C9orf72-associated G4C2 repeat RNA affects endogenous RuvBL1/2 levels is important. In line with this endogenous RuvBL1 and RuvBL2 levels were quantified in cell lysates from Figure 1 where cells were transfected with V5-45xG4C2 RAN producing DPR plasmids. The presence of RAN translated DPRs did not impact endogenous RuvBL1 and RuvBL2 protein levels. This new data is shown in Supplementary Figure 5 and is referred to on Page 16, Line 455 of the revised manuscript. Endogenous RuvBL1 and RuvBL2 levels are already shown for the C9orf72 patient cells used in Figure 2. The C9-500 BAC mice harbour approximately 500x G4C2 pathogenic repeats on their artificial chromosome and so we also quantified the endogenous RuvBL1 and RuvBL2 levels in the primary C9-500 BAC cortical neurons used in Figure 3. These new data are shown in Supplementary Figure 7. When compared to the non-transgenic controls the presence of the pathogenic C9orf72 repeat expansion did not significantly affect endogenous RuvBL1 levels. However, we did observe a significant reduction in total RuvBL2 levels in these transgenic cortical neurons compared to controls. This is described on Page 18 Line 502 of the revised manuscript. Given that there are only 45x G4C2 repeats in our C9-RAN reporter construct, these data suggest that it is the presence of a larger pathogenic repeat that has the potential to impact endogenous RuvBL2 levels. We did not quantify endogenous Pontin/Reptin levels in our *Drosophila* lines as these are DPR-only models and do not harbour the HRE. Thus, there was no repeat RNA expression.

Lastly, the introduction/discussion section focused on protein disaggregation/clearance properties of RuvBL1/2. However, their results clearly show most prominent effect of RuvBL1/2 overexpression is on RNA expression levels (transcription/metabolism including NMD); the authors should introduce/discuss more details about the effect of RuvBL1/2 at the RNA level.

We thank Reviewer 2 for their suggestion. We have now included substantial discussion on the role of nonsense mediated decay in relation to the C9orf72 repeat expansion and how the effect of RuvBL1/2 overexpression on repeat containing transcripts could relate to this. This discussion begins on Page 26 Line 755.

Minor points

The manuscript contains inaccurate generalizations (oversimplification) in various parts that overexpression of RuvBL1/2 reduces DPR even when the data do not support this.

Thank you. We have amended all inaccurate generalizations identified.

p8. last line. 2 μ DMH1 : Correction needed.

Thank you for indicating this omission. The text has now been updated to indicate the correct concentration (2 μ M).

p17 line 14-16 "Similar to what was seen with LV-FLAG-RuvBL1 transduction in primary cortical neurons, overexpression of RuvBL1 had no impact on poly(GA) or poly(GP) levels in these experiments." This statement is incorrect. Fig 3D shows a significant reduction of GA DPR upon RuvBL1 overexpression.

Apologies for this confusing statement. This sentence was indeed an inaccurate generalisation, and Reviewer 2 is of course correct; poly(GA) levels were reduced by RuvBL1 overexpression in primary cortical neurons. This sentence has now been updated for accuracy on Page 18, Line 517 of the revised manuscript.

p17-18 and Figure 5: Expressions of the gene names of Drosophila orthologs are inconsistent. In the text, it is described as RuvBL2, but in Fig. it is marked as Reptin.

Thank you for identifying this inconsistency. We have updated the text to use the correct Drosophila ortholog names whenever these are being discussed.

p18 line 2 Inconsistency of expression p18 line 2 "poly(PA)" Fig 5D "AP"

Thank you for pointing this out. We apologies for this inconsistency. In all cases the manuscript has been updated and now uses poly(PA) only.

p20-21 In the experiments in Figure 7F, G (RPL10A and RuvB1/2 co-IP experiments), there is no evidence that the overexpressed RPL10A is actually incorporated into the translational ribosome. So the reviewer thinks the authors cannot claim that "RuvBL1/2 were able to interact with the translational machinery" based solely on this experiment. We thank Reviewer 2 for their comment. We agree that this co-immunoprecipitation assay alone is insufficient to suggest an interaction with the translating ribosome. We have therefore chosen to tone down the language used to reflect this (Page 22 Paragraph beginning Line 623).

Figure 1: Endogenous expression levels of RuvBL1/2 need to be shown.

We have determined the effect of the repeat RNA on endogenous RuvBL1/2 levels in Supplementary Figure 5 and 7. However, in line with this suggestion we have re-run samples from Figure 1 and blotted with endogenous RuvBL1/2 antibodies. Endogenous RuvBL1/2 levels are now shown in panels A-E of Supplementary Figure 4 and are described on Page 15-16 Lines 429-437.

Figure 2 D, E, F and Figure 8 B, C, D lack each data point

Apologies for this oversight. The individual data points have been added to the graphs.

Figure 2 legend (D-F). Is one-way ANOVA with Tukey post-test correct?

Thank you for pointing out this error. Reviewer 2 is indeed correct, statistical differences were not assessed by Tukey post-test but were assessed by a student's t-test in these assays. The Figure legend has been updated for Figure 2.

Figure 3A-C untreated: untreated? Correction needed.

Thank you for pointing out this mistake. The Figure has now been updated.

Figure 6C Why the numbers (data points) of GR1000 mKate flies are small?

These are the numbers that were obtained from 3 independent biological crosses.

Figure 7A cyclin D blot signals are too weak.

These samples have now been re-run in order to better visualise the levels of cyclin D.

References

- 1 Castelli, L. M. *et al.* A cell-penetrant peptide blocking C9ORF72-repeat RNA nuclear export reduces the neurotoxic effects of dipeptide repeat proteins. *Sci Transl Med* **15**, eabo3823, doi:10.1126/scitranslmed.abo3823 (2023).
- 2 Bauer, C. S. *et al.* Loss of TMEM106B exacerbates C9ALS/FTD DPR pathology by disrupting autophagosome maturation. *Front Cell Neurosci* **16**, 1061559, doi:10.3389/fncel.2022.1061559 (2022).
- 3 Marchi, P. M. *et al.* C9ORF72-derived poly-GA DPRs undergo endocytic uptake in iAstrocytes and spread to motor neurons. *Life Sci Alliance* **5**, doi:10.26508/lsa.202101276 (2022).
- 4 Castelli, L. M. *et al.* SRSF1-dependent inhibition of C9ORF72-repeat RNA nuclear export: genome-wide mechanisms for neuroprotection in amyotrophic lateral sclerosis. *Mol Neurodegener* **16**, 53, doi:10.1186/s13024-021-00475-y (2021).
- 5 Webster, C. P. *et al.* The C9orf72 protein interacts with Rab1a and the ULK1 complex to regulate initiation of autophagy. *The EMBO Journal* **35**, 1656-1627, doi:10.15252/embj.201694401 (2016).
- 6 Gendron, T. F. *et al.* Cerebellar c9RAN proteins associate with clinical and neuropathological characteristics of C9ORF72 repeat expansion carriers. *Acta Neuropathol* **130**, 559-573, doi:10.1007/s00401-015-1474-4 (2015).
- 7 Quaegebeur, A., Glaria, I., Lashley, T. & Isaacs, A. M. Soluble and insoluble dipeptide repeat protein measurements in C9orf72-frontotemporal dementia brains show regional differential solubility and correlation of poly-GR with clinical severity. *Acta Neuropathol Commun* **8**, 184, doi:10.1186/s40478-020-01036-y (2020).
- 8 Lechtzier, V. *et al.* Sodium dodecyl sulphate-treated proteins as ligands in ELISA. *J Immunol Methods* **270**, 19-26, doi:10.1016/s0022-1759(02)00214-4 (2002).
- 9 Salomonsson, S. E. *et al.* Validated assays for the quantification of C9orf72 human pathology. *Sci Rep* **14**, 828, doi:10.1038/s41598-023-50667-3 (2024).
- 10 West, R. J. H. *et al.* Co-expression of C9orf72 related dipeptide-repeats over 1000 repeat units reveals age- and combination-specific phenotypic profiles in *Drosophila*. *Acta Neuropathol Commun* **8**, 158, doi:10.1186/s40478-020-01028-y (2020).
- 11 Freibaum, B. D. *et al.* GGGGCC repeat expansion in C9orf72 compromises nucleocytoplasmic transport. *Nature* **525**, 129-133, doi:10.1038/nature14974 (2015).

November 5, 2024

RE: Life Science Alliance Manuscript #LSA-2024-02757-TR

Dr. Christopher P. Webster
University of Sheffield
Neuroscience
SITRAN
385A Glossop Road
Sheffield S10 2HQ
United Kingdom

Dear Dr. Webster,

Thank you for submitting your revised manuscript entitled "RuvBL1/2 reduce toxic dipeptide-repeat protein burden in multiple models of C9orf72-ALS/FTD". We would be happy to publish your paper in Life Science Alliance pending final revisions necessary to meet our formatting guidelines.

- please address the Reviewer's remaining comments
- please be sure that the authorship listing and order is correct
- please make sure that the author order in the manuscript matches with the order entered in our system
- please consult our manuscript preparation guidelines <https://www.life-science-alliance.org/manuscript-prep> and make sure your manuscript sections are in the correct order
- please use the [10 author names, et al.] format in your references (i.e. limit the author names to the first 10)

A. FINAL FILES:

B. MANUSCRIPT ORGANIZATION AND FORMATTING:

Sincerely,

Reviewer #2 (Comments to the Authors (Required)):

I believe the authors have made generally favorable revisions. Especially, the newly added EGFP experiment strongly supports that the effect of RuvBL1/2 overexpression is specific to G4C2 RNA. Moreover, I highly appreciate the addition of the RuvBL1/2 knockdown experiment to clarify the results.

However, I would like to see an additional discussion on how the authors interpret the lack of effect of RuvBL1/2 knockdown on DPR and repeat RNA expression levels. If RuvBL1/2 is indeed associated with NMD, I assume that knocking RuvBL1/2 down will reduce NMD efficiency and resulting in increased expression levels of repeat RNA and DPR, but this was not the case.

In Legend for Figure 4A, it would be better to spell out abbreviations. For example, RA: Retinoic acid.

Supplementary figure 2B: In contrast to the result of figure 1E, the expression of V5-45xG4C2 seems not reduced upon overexpression of RuvBL1/2 (dot blot). It's a bit confusing. What is the difference?

Supplementary figure 9D does not display statistical test results. (ns?)

November 25, 2024

RE: Life Science Alliance Manuscript #LSA-2024-02757-TRR

Dr. Christopher P. Webster
University of Sheffield
Neuroscience
SITRAN
385A Glossop Road
Sheffield S10 2HQ
United Kingdom

Dear Dr. Webster,

Thank you for submitting your Research Article entitled "RuvBL1/2 reduce toxic dipeptide-repeat protein burden in multiple models of C9orf72-ALS/FTD". It is a pleasure to let you know that your manuscript is now accepted for publication in Life Science Alliance. Congratulations on this interesting work.

DISTRIBUTION OF MATERIALS:

Again, congratulations on a very nice paper. I hope you found the review process to be constructive and are pleased with how the manuscript was handled editorially. We look forward to future exciting submissions from your lab.

Sincerely,
